# Induced Model Matching:
# Restricted Models Help Train Full-Featured Models

**Usama Muneeb**
Electrical and Computer Engineering
University of Illinois Chicago
umunee2@uic.edu

**Mesrob I. Ohannessian**
Electrical and Computer Engineering
University of Illinois Chicago
mesrob@uic.edu

## Abstract

We consider scenarios where a very accurate (often small) predictive model using restricted features is available when training a full-featured (often larger) model. This restricted model may be thought of as "side-information", and can come either from an auxiliary dataset or from the same dataset by forcing the restriction. How can the restricted model be useful to the full model? To answer this, we introduce a methodology called Induced Model Matching (IMM). IMM aligns the context-restricted, or induced, version of the large model with the restricted model. We relate IMM to approaches such as noising, which is implicit in addressing the problem, and reverse knowledge distillation from weak teachers, which is explicit but does not exploit restriction being the nature of the weakness. We show that these prior methods can be thought of as approximations to IMM and can be problematic in terms of consistency. Experimentally, we first motivate IMM using logistic regression as a toy example. We then explore it in language modeling, the application that initially inspired it, and demonstrate it on both LSTM and transformer full models, using bigrams as restricted models. We lastly give a simple RL example, which shows that POMDP policies can help learn better MDP policies. The IMM principle is thus generally applicable in common scenarios where restricted data is cheaper to collect or restricted models are easier to learn.

## 1 Introduction

In many applications, it is both statistically and computationally easier to construct (often small) feature-restricted models. In this paper, we address the question of whether this could be beneficial in the construction of an (often larger) full-featured model.

To motivate, consider the following toy logistic regression problem, with further details in Section 7. Say we have three features $(x_1, x_2, x_3) \in \mathbb{R}^3$ and a binary label $y \in \{0, 1\}$. Let features be generated from a distribution $\pi$ and let labels be conditionally generated according to a logistic (full-featured) *true model* $P(y|x_1, x_2, x_3)$. Assume we have ample feature-restricted data with only one feature $(x_1, y)$, which, by marginalization, are samples from the (feature-restricted) *true induced model* $\overline{P}(y|x_1) = \sum_{x_2, x_3} \pi(x_2, x_3|x_1) P(y|x_1, x_2, x_3)$. By virtue of this data, we get a very good approximation $\hat{P}$ of $\overline{P}$, which we can use as a *target induced model*. **How do we use $\hat{P}$, along with full-featured data $(x_1, x_2, x_3, y)$ to obtain a (full-featured) *learned model* $Q(y|x_1, x_2, x_3)$?** A few possible approaches are as follows. We could ignore $\hat{P}$, and learn $Q$ simply by minimizing cross-entropy on the data. This is wasteful, because $\hat{P}$ contains valuable information. Alternatively, we could learn $Q$ by minimizing cross-entropy in addition to a secondary loss that keeps $Q$ close to $\hat{P}$. This is reasonable — in Section 2 we relate this to reverse knowledge distillation and in Section 6 to noising. However, $\hat{P}$ addresses a markedly different task than $Q$, i.e., that of predicting with a

38th Conference on Neural Information Processing Systems (NeurIPS 2024).

restricted set of features. Instead, what this paper proposes is to equalize the field when comparing to $\hat{P}$, by inducing a restricted model $\overline{Q}(y|x_1)$ from $Q$ during training, and using a secondary loss to match $\hat{P}$ to $\overline{Q}$, rather than to $Q$. We call this *induced model matching* (IMM). In Figure 1 we show how this speeds up learning and reduces predictor variance.

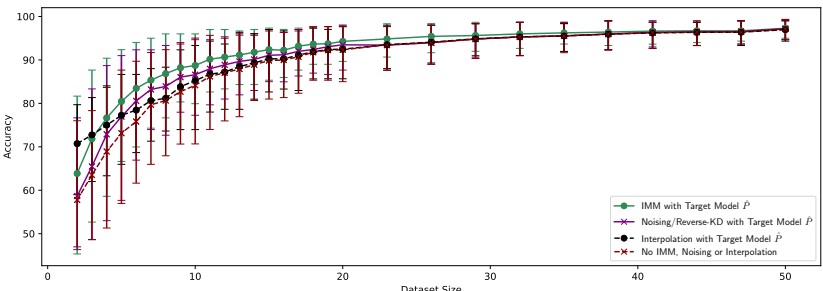

*Figure 1: Comparing test accuracy of logistic model, trained using interpolation, noising, and IMM (with a Bayes Optimal $\overline{P}$). Bars are $10^{th}$ to $90^{th}$ percentiles of $300$ runs.*

This example gives an overview of the entire process behind IMM. Most of the paper, however, is dedicated to language modeling, where very natural restricted models exist, namely $N$-grams. The inspiration of this research is in fact rooted in certain language model data augmentation techniques that *noise* the data using $N$-grams Xie et al. (2017). The fresh perspective that we offer here is that such noising is best understood as an attempt to incorporate a feature-restricted model's knowledge into the full-featured model being trained. This interpretation reveals the general fundamental question: **If we are armed with a very accurate feature-restricted model, what is the right way to incorporate it into the training of a model with a larger feature set?**

Our contributions and organization are as follows:

1. In Section 3, we rigorously frame this question through the notion of an *induced model*.

2. In Section 4, we use this framework to propose a strategy to incorporate the knowledge of the restricted model during learning. This consists of using a regularizer that matches the predictions of the induced learned model with that of the desired target restricted model. This is the language model instance of the *induced model matching* (IMM) methodology.

3. In Section 5, we propose computational approaches to make IMM practical and pave the way for further scaling IMM.

4. In Section 6, we loop back to relate IMM to the noising approach of Xie et al. (2017) and to reverse knowledge distillation. We share two key findings. First, these alternatives may be thought of as approximations to IMM. Second, they have a major caveat. They may not be consistent even in the ideal infinite-data regime, in contrast to IMM, which is consistent.

5. In Section 7, we experimentally demonstrate the effectiveness of IMM through: (1) details of the logistic regression example, (2) experiments on an LSTM RNN for language modeling and on BERT for classification, showing improvements on multiple tasks, and (3) a simple reinforcement learning example that illustrates the further potential of IMM.

In Section 8 we discuss limitations and applications beyond those illustrated in the paper.

## 2    Related Works

We review two lines of work that are closely related to IMM, noising and knowledge distillation, which are respectively implicit and explicit versions of the same idea. We also review key literature showing the continued merit of $N$-grams as restricted models.

**Noising and Data Augmentation in Language Modeling**    The noising methodology in language modeling was proposed initially in order to provide a Natural Language Processing (NLP) parallel to the many data augmentation techniques that existed in other machine learning applications (e.g., sampling translated and rotated variants, in image processing and computer vision).

Most data augmentation techniques in NLP have used noising or smoothing in one form or another. Feature noising by Wang et al. (2013) was one of the earliest attempts at noising when it came to structured prediction tasks. Later, Xie et al. (2017) successfully used restricted language models, namely unigrams and bigrams, to perform data augmentation through noising. By making a rough connection between noising and smoothing, Xie et al. (2017) were able to show a clear advantage, beyond the use of other regularization techniques such as dropout. Section 6 more closely examines the claims of that paper and connects it to the framework of the present one.

The current approach is not related to *all* data augmentation techniques, rather specifically to those that use noising with a restricted model. Indeed, it can complement other data augmentation techniques. For example, in our BERT experiments, this technique complements the MLM objective, where the masking itself can be thought of as data augmentation. Similarly, it is capable of complementing alternative masking and corruption techniques, such as EDA (Wei and Zou, 2019) and SSMBA (Ng et al., 2020), which are known to outperform BERT's default masking for the MLM objective. Similarly, IMM's gains are in addition to those from other regularization techniques, e.g., weight decay and dropout (Srivastava et al., 2014), as these can be simultaneously utilized.

**$N$-grams and their Merits**   $N$-grams are based on co-occurrence counts, which makes them easy to learn but limits their usefulness for long histories. However, common techniques such as smoothing and backoff make it possible to build excellent short-context models such as bigrams and trigrams, which can rival or exceed neural models that use the same context size (Chelba et al., 2017). This hints at there being value in using these $N$-gram models to improve long-context models. Some of the earliest attempts at this interpolate the output of the smaller model with modern models. The continued relevance of this approach is evidenced in a very recent paper (Liu et al., 2024) that proposes a special data structure to precompute $N$-grams to arbitrary lengths and uses interpolation to improve larger language models (LMs). A key motivator of the current paper is the approach of Xie et al. (2017), which instead noises the training data of LMs using $N$-grams, with improved outcomes over interpolation. Note that other approaches that take advantage of $N$-grams also exist, such as that of Li et al. (2022), who let LMs learn what the $N$-gram could not, i.e., the residual.

**Knowledge Distillation**   Knowledge distillation (KD) is a paradigm first proposed to let a powerful teacher model help better train a weaker student model, by complementing the hard labels of existing data with soft labels (Hinton et al., 2015). In the present notation, the resulting average loss takes the following general form,

$$\mathsf{Cross\text{-}Entropy}\,(Q) + \lambda \sum_x \pi_n(x) \mathsf{D}_{\mathsf{KL}}\left(P_\tau^{\mathsf{teacher}}(\cdot|x)\middle\|Q_\tau(\cdot|x)\right), \tag{1}$$

where $\mathsf{Cross\text{-}Entropy}$ uses hard data, $x$ are contexts, $\pi_n$ is the empirical distribution of contexts, $Q$ is the learned/student prediction model, and $P^{\mathsf{teacher}}$ is the teacher model, which can be thought of as providing soft predictions. $\tau$, the softmax temperature, is used for smoothing the outputs.

Most relevant to IMM is the recent discovery that, paradoxically, KD with a weak teacher can be helpful to a powerful student. This "reverse-KD" or "distillation from weak teachers" phenomenon was first demonstrated in vision by Yuan et al. (2020), who interpret and ascribe its performance to smoothing targets in a context-dependent way using the teacher, in contrast to typical label smoothing that can be interpreted as using a uniform distribution. Though this comes years after the noising papers, it parallels closely how Xie et al. (2017) transition from uniform noising to Kneser–Ney noising. Later, reverse-KD was also shown to be effective in language models (Qin et al., 2021).

On the surface, IMM is similar to reverse-KD, as it uses a weak target teacher to regularize a powerful student's learning. A quick comparison to Eq. (14) when $P^{\mathsf{teacher}}$ is the restricted model $\overline{P}$ reveals that reverse-KD has more in common with noising, which we show can be sub-optimal. Indeed, the major difference between reverse-KD and IMM is the fact that *the weakness of the teacher in this case is of a very particular nature* — it stems from the reliance on a restricted context. Reverse-KD ignores this fact by comparing the student model $Q$ *directly* to the teacher model in the KL term of Eq. (1). In contrast, IMM harnesses this fact, by comparing the student *indirectly* to the teacher (target model) at its own level, i.e., using the induced model $\overline{Q}$. Very recent work by Lee et al. (2023) shows that reverse-KD in language, unlike in vision, can be harmful with very weak teachers, and thus it is not advisable to use reverse-KD with the simple restricted models, e.g., bigrams, that we consider here. Our work shows that they *can* be effectively incorporated, if we use IMM instead.

## 3 Problem Description

**Problem Setting**  We consider the problem of training a full model $Q(y_t|x_t)$ that is able to predict a label of data point $t$ based on its *full context*. For example, in forward predictive models, $x_t$ is the sequence of past tokens and $y_t$ is the next token. Throughout the paper, $Q$ always denotes the full model and is deemed to belong to a reasonably large functional class. Given context-prediction data $(x_t, y_t)$ of size $n$, possibly tokenized from a single text, let the primary loss used for training this model be the log-loss. Thus training minimizes the cross-entropy risk:

$$\mathsf{Cross\text{-}Entropy}(Q) = -\sum_t \log Q(y_t|x_t) \equiv \sum_x \pi_n(x) \sum_y P_n(y|x) \log \frac{1}{Q(y|x)},$$

where $\pi_n$ is the empirical distribution of the context and $P_n$ be the empirical distribution of the prediction given the context. Note that $(x_t, y_t)$ refer to tokens while $(x, y)$ refer to types. Minimizing this empirical risk can be seen as $Q$ striving to approach the "true" model $P(y_t|x_t)$ generating the data, in average context-conditional KL divergence. The idealized risk is thus:

$$\sum_x \pi(x) \sum_y P(y|x) \log \frac{P(y|x)}{Q(y|x)} \equiv \sum_x \pi(x) \, \mathsf{D_{KL}} \left( P(\cdot|x) \| Q(\cdot|x) \right) \tag{2}$$

where now $\pi$ is also the true (not empirical) *context distribution*.

**Induced Models**  While $Q$ strives to approximate all of $P$, we may have additional knowledge about $P$ that captures some of its properties. We consider in particular knowledge of the following form. Assume that the full context $x_t$ can be decomposed into a short context $\overline{x}_t$ and an extended context $\underline{x}_t$, and that one has access to the "true" model that predicts $y_t$ based solely on the short context $\overline{x}_t$, i.e. $\overline{P}(y|\overline{x})$. To make the notation easier to follow, we invite the reader to consult the glossary of notations in Appendix A. How is this restricted model related to $P$ and $\pi$? By a simple marginalization argument, we can see that this model is:

$$\overline{P}(y|\overline{x}) = \sum_{\underline{x}} \mathbf{P}(y, \underline{x}|\overline{x}) = \sum_{\underline{x}} \pi(\underline{x}|\overline{x}) \, P(y|\overline{x}, \underline{x}) \tag{3}$$

We call $\overline{P}$ the true *induced model*. It depends both on the context distribution $\pi$ and the true model $P$. Since we do not have the latter two, we cannot explicitly compute $\overline{P}$. What motivates us, however, is the possibility to learn it more accurately than $P$, either by virtue of its smaller parametrization or thanks to cheaply procured auxiliary context-restricted data.

**Problem Statement**  Given knowledge of the true induced model $\overline{P}$, or a very good approximation thereof, how can this information be incorporated to improve the learned model $Q$?

## 4 Induced Model Matching (IMM)

**Construction of the IMM risk**  To address the problem, we introduce a secondary loss that *matches* the learned model's prediction with that of the *induced model*, whence the name *Induced Model Matching* or IMM. The key insight here is *not* to match $\overline{P}$ with $Q$, but rather with $\overline{Q}$, the learned induced model that, just like the move from $P$ to $\overline{P}$ in Eq. (3), specializes $Q$ to performing predictions with only the short context:

$$\overline{Q}(y|\overline{x}) = \sum_{\underline{x}} \pi(\underline{x}|\overline{x}) Q(y|\overline{x}, \underline{x}) \tag{4}$$

Let's first idealize and assume availability of $\overline{P}$ and the context distribution $\pi$, required to compute $\overline{Q}$. Equipped with $\overline{Q}$, we can introduce the **idealized** *induced model matching* (IMM) risk, a secondary risk that is the average context-conditional KL divergence with the restricted context:

$$\sum_x \pi(x) \underbrace{\sum_y \overline{P}(y|\overline{x}) \log \frac{\overline{P}(y|\overline{x})}{\overline{Q}(y|\overline{x})}}_{\mathsf{D_{KL}}(\overline{P}(\cdot|\overline{x}) \| \overline{Q}(\cdot|\overline{x}))} \tag{5}$$

Since $\overline{P}$ and $\pi$ are not available in practice, the idealized IMM risk cannot be computed. However, as the core motivation of using $\overline{P}$ is the potential ability to learn it very accurately from data, we assume instead that we have access to a *target induced model* $\hat{P}$ as a proxy to $\overline{P}$. As for calculating $\overline{Q}$, knowledge of $\pi$ in Eq. (4) can be intuitively understood as a mechanism for filling-in for the extended context, based on the short context. As such, we have the following natural empirical version which can be thought of as averaging $Q$ over all extended contexts in the dataset, while keeping the short context fixed:

$$\hat{Q}(y|\overline{x}) \propto \sum_t \mathbf{1}\{\overline{x}_t = \overline{x}\}Q(y|x_t) \tag{6}$$

The proportionality constant is unimportant as, thanks to the logarithm, it contributes only a constant to the overall risk. By combining these empirical proxies, we obtain **the empirical IMM risk**:

$$\mathsf{IMM}(Q) = \sum_x \pi_n(x) \sum_y \hat{P}(y|\overline{x}) \log \frac{1}{\hat{Q}(y|\overline{x})} \tag{7}$$

This mirrors Eq. (5), up to $\overline{P}$ inside the logarithm, which only contributes an additive entropy term that does not depend on $Q$ and is thus irrelevant to optimization.

**IMM as a regularizer**  Given a reliable $\hat{P}$, we propose to incorporate this knowledge into the full model by using the IMM risk as a regularizer. Using a hyper-parameter $\lambda$ that can be tuned, our *induced model matching* methodology consists of training the model $Q$ by minimizing

$$\mathsf{Cross\text{-}Entropy}(Q) + \lambda\,\mathsf{IMM}(Q). \tag{8}$$

**Separating IMM into components**  To weave IMM into existing ML optimization pipelines, it is useful to treat it as a separable risk. For this, we can rewrite Eq. (7) by expanding the empirical distribution $\pi_n$. Then, the components of this separable risk are given by the conditional cross-entropies between $\hat{P}$ and $\hat{Q}$, because we can write:

$$\mathsf{IMM}(Q) = -\frac{1}{n}\sum_t \underbrace{\left[\sum_y \hat{P}(y|\overline{x}_t) \log \hat{Q}(y|\overline{x}_t)\right]}_{\mathsf{IMM}_t(Q)}. \tag{9}$$

The simplest version of $\hat{P}$ could be based on empirical counts, which may be valid if the context space is discrete and small. In language modeling, smoothing methods can be used to generate better versions of $\hat{P}$, such as the modified Kneser–Ney smoothed version of the bigram counts (Kneser and Ney, 1995; Chen and Goodman, 1999) that can be expressed as

$$\hat{P}(y|\overline{x}) = \frac{1}{n}\sum_t (1 - \nu(\overline{x}))\mathbf{1}\{y_t = y\} + \nu(\overline{x})b(y),$$

where $\nu(\overline{x})$ is the missing mass for previous token $\overline{x}$ and $b$ is the back-off distribution. For logistic regression in Section 7 the context space is continuous, and these components come from a separately trained one-feature predictor. For our RL example, they come from the optimal actions of a POMDP.

## 5  Computation

A direct implementation of IMM is prohibitive, since evaluation of the risk, Eq. (8), requires making a secondary pass over the entire dataset for each $t$, when inducing the model in Eq. (6). We address this with two approaches. The first approximates the objective by replacing this secondary pass with *sampling* whereas the second incorporates this secondary pass into the primary pass by *serializing* the gradient calculation. *Sampled IMM* has the advantage of low gradient variance at the expense of added computation. *Serialized IMM* has higher gradient variance but has only constant-factor computational overhead. All of our experiments use sampled IMM, with the exception of a demonstration of serialized IMM for logistic regression (in Appendix D.5), as a proof of concept for IMM's scalability.

**Sampled IMM**   To alleviate the exact computation of IMM, we could maintain a dictionary/multiset of extended contexts for each short context[1], and sweeping only across those. This amounts to rewriting Eq. (6) as follows:

$$\hat{Q}(y|\overline{x}) \propto \sum_{t' \,:\, \overline{x}_{t'}=\overline{x}_t} Q(y|\overline{x}_t, \underline{x}_{t'}) = \sum_{x \in \mathsf{extend}(\overline{x}_t)} Q(y|\overline{x}_t, x), \text{where } \mathsf{extend}(\overline{x}) = \biguplus_{t \,:\, \overline{x}_t=\overline{x}} \{\!\{\underline{x}_t\}\!\} \quad (10)$$

is the multiset of extended contexts of the short context of $x$.

Since a given short context may appear in a large number of extended contexts, the dictionary/multiset approach of Eq. (10) remains expensive. In particular, gradients need to be computed with each of these contexts. Instead, we propose an approximation based on writing the (normalized) sum as an expectation over samples from $\mathsf{extend}(\overline{x}_t)$, followed by approximating this expectation with $k$ samples. As a result, the IMM component at $t$ becomes:

$$\mathsf{IMM}_t(Q) = -\sum_y \hat{P}(y|\overline{x}) \log\left[\mathbf{E}_{X \sim \mathsf{extend}(\overline{x}_t)}\left[Q(y|\overline{x}_t, X)\right]\right] \quad (11)$$

$$\approx -\sum_y \hat{P}(y|\overline{x}) \log\left[\frac{1}{k}\sum_{i=1}^k Q(y|\overline{x}_t, x_i)\right], \quad (12)$$

where $x_i \sim \mathsf{extend}(\overline{x}_t)$ are samples from the multiset of extended contexts. Implementation details are deferred to Appendix C, including Algorithms 1 and 2 that describe a typical training loop when sampled IMM or serialized IMM are incorporated into SGD respectively.

At first glance, it appears that sampled IMM requires maintaining $k$ copies of the model. However, this can be sequentialized, as explained in Appendix C.2. This means that the space overhead during training is a factor of 2 compared to the baseline of no-IMM (i.e., a second set of gradients). However, the time overhead is generally a $k$-fold increase over the baseline, but can be worse for recurrent models such as LSTMs. This is because we typically unroll the model over $L$ tokens and can perform forward/backward passes over this unroll, in $L$ steps. If we apply IMM at every unroll position, we need to restart the LSTM, which then incurs an $\mathcal{O}(kL)$-fold increase. We partially mitigate this in our current implementation by applying IMM periodically (not at every iteration), see Appendix D.2.2. For example, if we apply it only every $\Omega(1/L)$ iterations, the overhead factor remains $\mathcal{O}(k)$.

**Serialized IMM**   Sampled IMM delivers the benefits of IMM, but at a computational cost. To make IMM truly scalable, it is imperative to bring it to near-constant factor overhead to the baseline of no-IMM. The main bottleneck is the calculation of the learned induced model $\hat{Q}$. The following alternative approach bears a resemblance to the *sequentialization* aspect covered in Appendix C.2, Eq. (20). Up to a constant, we can write the idealized IMM risk of Eq. (5) as an averaging only over short contexts:

$$-\sum_{\overline{x}} \pi(\overline{x}) \sum_y \overline{P}(y|\overline{x}) \log \overline{Q}(y|\overline{x}), \text{ where the induced model is } \overline{Q}(y|\overline{x}) = \sum_{\underline{x}} \pi(\underline{x}|\overline{x}) Q(y|\underline{x}, \overline{x}).$$

The gradient of this IMM risk then becomes:

$$-\sum_{\overline{x}} \pi(\overline{x}) \sum_y \overline{P}(y|\overline{x}) \frac{\sum_{\underline{x}} \pi(\underline{x}|\overline{x}) \nabla Q(y|\underline{x}, \overline{x})}{\overline{Q}(y|\overline{x})} = -\sum_{\overline{x}} \sum_{\underline{x}} \pi(\overline{x}) \pi(\underline{x}|\overline{x}) \sum_y \overline{P}(y|\overline{x}) \frac{\nabla Q(y|\underline{x}, \overline{x})}{\overline{Q}(y|\overline{x})}$$

The empirical version of this gradient is:

$$-\frac{1}{n}\sum_t \sum_y \hat{P}(y|\overline{x}_t) \frac{\nabla Q(y|\underline{x}_t, \overline{x}_t)}{\hat{Q}(y|\overline{x}_t)} = \frac{1}{n}\sum_t \nabla\overbrace{\left(-\sum_y \overline{P}(y|\overline{x}_t) \underbrace{\frac{Q(y|\underline{x}_t, \overline{x}_t)}{\hat{Q}(y|\overline{x}_t)}}_{\textbf{correction}} \log Q(y|\underline{x}_t, \overline{x}_t)\right)}^{\widetilde{\mathsf{IMM}}_t} \quad (13)$$

The last expression in Eq. (13) is in the form of a "corrected" cross-entropy between $\hat{P}$ and $\hat{Q}$, which we denote by $\widetilde{\mathsf{IMM}}_t$. The correction factor is **not differentiated** and is the ratio or Radon–Nikodym

---

[1]This requires space $\mathcal{O}(n \log n)$ only, because each key in the dictionary is a short context, and each value is a set of pointers to the data, with set size equal to the number of times that short context appears. Adding up the number of occurrences of all short histories gives us $n$ and each pointer requires $\mathcal{O}(\log n)$ bits.

derivative of $Q$ relative to $\hat{Q}$. What this accomplishes is to delegate the averaging of the gradients to the primary pass over the dataset. If the correction term is given, computing this gradient costs the same as computing the baseline of no-IMM gradients. There are two caveats: first, because the averaging is now happening in the primary pass, the variance of this gradient is higher than sampled IMM and, second, the correction factor still requires knowing the learned induced model $\hat{Q}$. To address the higher variance of the gradients, techniques such as momentum approaches along with learning rate schedulers can be used. To address knowing $\hat{Q}$, we suggest the heuristic of updating $\hat{Q}$ only periodically. Then, if model $Q_\dagger$ (that eventually becomes stale) is used to calculate $\hat{Q}_\dagger$, use the ratio $Q_\dagger/\hat{Q}_\dagger$ for correction factor. Using the current $Q$ tends to cause instability, likely due to the correction no longer obeying expected constraints, e.g., mean 1 for every $y$. By choosing the update period inversely proportionally to the cost of updating $\hat{Q}_\dagger$ and by keeping $Q_\dagger$ in memory, we incur an $\mathcal{O}(1)$ factor increase in time and space compared to no-IMM, which makes this variant of IMM highly scalable. Note that without the correction factor, IMM turns into reverse-KD and noising, a connection that we elaborate on in Section 6.

## 6 Analysis

In what follows, we assume to be in the realizable case, i.e., that the model space of $Q$ is expressive enough to contain the true model $P$. We also take an infinite-data perspective and focus on consistency only, even though IMM also confers a finite-sample advantage (understanding this advantage analytically is worth studying in the future.) We show (1) that IMM is consistent, (2) that noising and reverse-KD are akin to single-sample IMM, (3) that this shows that they minimize an upper bound on the IMM risk, and finally (4) that this introduces sub-optimality, which could even threaten consistency. This gives basic analytical backing to why IMM is preferable.

**Consistency of IMM** Observe that if, in the main objective, Eq. (8), cross-entropy and IMM were replaced with their true counterparts, Eqs. (2) and (5) respectively, then $Q = P$ remains the minimizer of the objective. This observation shows that we recover the true model in the infinite-data regime, i.e., that IMM is consistent for all $\lambda$. We next aim to show that the key caveat of noising and reverse-KD is that they may be inconsistent, unless $\lambda$ is made to vanish.

**From single-sample IMM to noising and reverse-KD** We first explain how IMM and noising are related. Experimentally, using a single sample ($k = 1$) in the IMM approximation of Eq. (12) produces perplexities that are near-identical to noising. We explain this phenomenon as follows. Say data point $t$ is considered during optimization, in an SGD mini-batch. When a single random extended context is sampled, it is equivalent to swapping that data point's context with another based on the multiset $\mathrm{extend}(\overline{x}_t)$. That context belongs to a different data point $t'$. Since sampling is done uniformly from the multiset, this simply presents data to the SGD algorithm in a different random order, possibly with repetition. More pertinently to noising, the actual prediction $y_{t'}$ has no role in calculating the loss. Instead, the prediction is randomized through the sum over all possible $y$ in Eq. (9). Though not identical, this is a very close map to the noising-based data augmentation proposed by Xie et al. (2017), namely prediction noising (see Appendix B.1 for a review of this framework and why prediction noising is its main aspect.) It is in fact even closer to an alternative proposed later by Gao et al. (2019), which uses soft prediction vectors just like $\hat{P}$ in this single-sample approximation of IMM. As a result of this argument, we can think of noising as minimizing the following idealized objective, which coincidentally is equivalent to the reverse-KD objective (Yuan et al., 2020; Qin et al., 2021) (see also Section 2) with $\overline{P}$ as the teacher:

$$\mathsf{Cross\text{-}Entropy}(Q) + \lambda \sum_x \pi(x) \sum_y \overline{P}(y|\overline{x}) \log \frac{1}{\underbrace{Q(y|x)}_{\text{key difference}}} \tag{14}$$

**Inconsistency of noising and reverse-KD** How is this single-sample IMM objective related to performing IMM? Recall that we can write a single component of the empirical IMM risk with sampling as in (12), which by Jensen's inequality can be upper bounded as follows:

$$-\sum_y \hat{P}(y|\overline{x}) \log \left( \mathbf{E}_X \left[ Q(y|\overline{x}_t, X) \right] \right) \leq -\sum_y \hat{P}(y|\overline{x}) \mathbf{E}_X \left[ \log \left( Q(y|\overline{x}_t, X) \right) \right] \tag{15}$$

A single-sample approximation of the expectation *inside* the $\log$ is in fact a biased estimator of the left-hand side. However, it is an unbiased estimator of the right-hand side, with the expectation *outside* of the $\log$. Thus, noising and reverse-KD upper bound the IMM risk. Minimizing an upper bound instead of the desired risk *could* introduce suboptimality. The following proposition uses the difference between these methods, which pit the target model against the full learned model $Q$ and *not* the induced learned model $\overline{Q}$ like IMM (contrast Eq. (14) and Eq. (5)), to show that this suboptimality can indeed occur. Even in the realizable case and with infinite data, IMM is always consistent but there exists a counterexample where noising and reverse-KD fail to consistently recover the true model. The proof is in Appendix B.2.

**Proposition 6.1.** *Assume that we are optimizing the idealized noising objective of Eq.* (14) *— i.e., we are operating in the infinite-data regime — and let $Q^\star$ be its global minimizer. Assume further that the model class for $Q$ contains the true model $P$ — i.e., we are in the realizable case. Then, there exists a choice of $\pi$ and $P$ such that $Q^\star \neq P$.*

## 7 Experimental Results

### 7.1 Starting Toy Example: Logistic Regression

Consider the logistic regression example from the introduction. The main results are given in Figure 1 and the full experimental details can be found in Appendix D.1. Here we highlight how the problem fits the IMM framework and where it deviates from language modeling. First, note that the context decomposition is $\overline{x} = x_1$ and $\underline{x} = (x_2, x_3)$.

**Setting** We sample features uniformly over a cube and assume we have ample data points of the form $(x_1, y)$. This allows us to build an excellent *restricted model* to predict the label based on just $x_1$, call it $\overline{P}(y|x_1)$, nearly close to the (restricted) Bayes predictor or true conditional probability. Just like in language modeling, to induce a model we need to draw $\underline{x}$'s from its conditional distribution given $\overline{x}$. $\overline{Q}(y|x_1)$, the *induced model* of $Q(y|x_1, x_2, x_3)$, can then be interpreted as the average of $Q$'s predictions, when $\underline{x}_t = (x_2, x_3)$ is drawn from its conditional distribution given $\overline{x}_t = x_1$. Since we typically don't have access to this distribution, we approximate it empirically. In language modeling, we could just sample from the empirical distribution of $\underline{x}$ for a given $\overline{x}$. In logistic regression, this is not viable since $x_1$ is continuous and does not repeat. We rely instead on density estimation. We use a soft nearest-neighbor density estimate $\hat{f}(x_2, x_3|x_1) \propto \sum_{t=1}^{n} \delta_{x_{2,t}, x_{3,t}}(x_2, x_3) e^{-\alpha|x_{1,t} - x_1|}$, where $1/\alpha$ is the bandwidth of the Laplace kernel. (With cross-validation, we determine $\alpha = 1$ to be a good choice.) If we let $w_t(x_1) = e^{-\alpha|x_{1,t} - x_1|}$, the resulting induced model by marginalization is:

$$\overline{Q}(y|\overline{x}) = \int f(\underline{x}|\overline{x}) Q(y|\overline{x}, \underline{x}) \approx \sum_{t=1}^{n} \frac{w_t(\overline{x})}{\sum_{t=1}^{n} w_t(\overline{x})} Q(y|\overline{x}, \underline{x}_t)$$

These equations are respectively equivalent to Eqs. (4) and (6). The IMM risk and the corresponding overall objective remain the same:

$$\mathsf{IMM}(Q) = \sum_{t=1}^{n} \sum_{y=0,1} \hat{P}(y|x_{1,t}) \log \frac{1}{\overline{Q}(y|x_{1,t})}, \qquad \mathsf{Cross\text{-}Entropy}(Q) + \lambda\, \mathsf{IMM}(Q).$$

**Results** In Figure 1, we compare the performance of IMM-trained $Q$ (green) to that without IMM (maroon). We sweep a range of $n$ from 2 to 50, and use a cross-validation optimized $\lambda$ for each (details in the Appendix D.1.1). The key observations are that: (1) IMM always improves on the baseline performance, (2) the variance of the outcomes is also typically diminished, (3) the improvement is greater with less data, but the gain across data sizes is equivalent to access to an average of $30\%$ extra data. This and similar experiments suggest that gains are highest when the dataset size is comparable to the number of parameters. This simple scenario demonstrates how IMM effectively harnesses the benefit of accurate feature-restricted models when training full-featured models. For reference, we also include the results of noising and interpolation. IMM is always better than noising, but interestingly interpolation is better with very few samples, though much worse with more. We attribute this to the fact that with less data it is harder to obtain an accurate induced model.

Table 1: *Perplexity for an LSTM Language Model using the Penn TreeBank dataset. The numbers on None and KN Noising are from Xie et al. (2017) and can be replicated using their original code (we use the model with latent dimension 1500). Like the baseline, for each row, we report the best value across as many restarts.*

| Improvement | Validation | Test |
|---|---|---|
| None (only regular dropout) | 81.6 | 77.5 |
| KN Noising (reproducible) | 76.7 | 73.9 |
| IMM with KN Bigram | **76.0** | **73.3** |

Table 2: *Results on the BERT$_{BASE}$ Language Model. The baseline numbers can be replicated using the original BERT code by Google, as well as our provided repository. Matthew's Correlation Coefficient is used for CoLA, F1 score for MRPC and Accuracy for QNLI and RTE. Like the baseline, reported numbers are averages across multiple restarts.*

| | BERT$_{BASE}$ | + MLM | +IMM |
|---|---|---|---|
| CoLA | $52.1 \pm 4.0$ | $55.0 \pm 3.0$ | $\mathbf{60.0 \pm 1.0}$ |
| MRPC | $88.9 \pm 2.0$ | $89 \pm 1.0$ | $\mathbf{90 \pm 1.0}$ |
| QNLI | $90.5 \pm 2.0$ | $91.0 \pm 2.0$ | $\mathbf{93.5 \pm 1.0}$ |
| RTE | $66 \pm 3.0$ | $68 \pm 2.0$ | $\mathbf{71 \pm 1.0}$ |

## 7.2 Language Modeling Experiments

In these language modeling experiments, the restricted model we use is the modified Kneser–Ney bigram (Kneser and Ney, 1995; Chen and Goodman, 1999) of the dataset in question. To see why this is a good choice, we refer the reader to benchmarking done by Chelba et al. (2017); neural models (single layer and 2-layer LSTM RNNs) could not improve upon the perplexity of an interpolated Kneser–Ney $N$-gram of a similar order. After the introduction of the attention mechanism (Bahdanau et al., 2014; Luong et al., 2015), better neural models now dominate language modeling (Vaswani et al., 2017; Devlin et al., 2018; Turc et al., 2019). We investigate how IMM could potentially improve even these more modern models, by using BERT's performance on some GLUE benchmarks as a proof of concept. (Appendix D.2.1 gives evidence that IMM indeed improves the full model's performance on the restricted task.)

**LSTM RNN Experiments**  We build on top of the code provided by Xie et al. (2017) using the same topology for the LSTM RNN. The chief purpose of these experiments is to contrast directly with noising introduced in that paper. The PTB dataset is a document dataset and the LLM is solving a *next word prediction* task. The average cross-entropy of multiple unroll positions of the RNN, or exponentiated as *perplexity*, is used as the measure of performance. For training and measuring validation perplexity, $L = 35$ unroll positions are used. During testing, only 1 unroll position is used. The IMM component is always calculated by running the LSTM in an evaluation mode (i.e., without any dropout). In addition, while regular evaluation updates the state in stateful models like the LSTM, the IMM branch never changes the state and uses the state set by the primary branch when traversing the dataset. In Table 1, we report perplexity values using $k$-sampled IMM with $k = 10$. The table also includes the best noising results that we could reproduce based on the code of Xie et al. (2017), after communication with the authors (these are 0.5 more than the paper's numbers).

**BERT Experiments**  We introduce the IMM objective in BERT's *fine-tuning* phase by reintroducing the Masked Language Model (MLM) objective that is originally only present during pre-training. In Google's original BERT code, MLM was present during pre-training but removed from fine-tuning, possibly because of minimal gain. For us, however, MLM is ideally suited to be augmented with the IMM objective because it is based on cross-entropy and we can similarly generate an *induced bigram* for predicting masked words in a sequence of tokens. We report numbers on these datasets in Table 2. The second column shows the numbers after adding back the MLM objective, which doesn't produce much gain on its own. The third column adds IMM within MLM, significantly boosting the gains.

Since some GLUE (Wang et al., 2018) datasets (used in the original BERT paper) are too large to be trained in an academic setting, we use a subset of the GLUE tasks (Warstadt et al., 2018; Dolan and Brockett, 2005; Rajpurkar et al., 2016; Dagan et al., 2005) to demonstrate the gains using IMM. For diversity, our selected GLUE tasks are a mixture of single-sentence and double-sentence tasks. Further experimental details are provided in Appendix D.2.

### 7.3 Reinforcement Learning: POMDPs helping MDPs

IMM is particularly appealing in situations where incomplete-feature data may be much more available, due to reasons like law, privacy, or limited sensing. If an autonomous driving system is developed with few sensors, and later a new system taking advantage of more sensors is to be designed, the older system may act as a restricted model helping the new design. If a diagnostic system is built based on a limited set of genetic markers, and later more markers are determined relevant, then the legacy system can be used without referring to the legacy data, which may need to be kept private. If a platform's recommendation and ad engine is trained from ample general-public data, and later a personalized engine is to be developed, then the older engine can inform the personalized one through IMM.

In stateful environments, problems like the latter often require a reinforcement-learning (RL) solution. If the personalized engine has full-featured data, it can use the data to train an MDP (Markov Decision Process) policy to optimize expected reward (Sutton and Barto, 2018). In contrast, the general-public data, despite being abundant, may lack in features and may only allow for solving a POMDP (Partially Observable Markov Decision Process). We can show that IMM can allow a good POMDP solution to significantly improve MDP training, by modifying policy-gradient methods such as REINFORCE (Williams, 1992). In Figure 2, we illustrate the reward achieved with and without the use of IMM, for learning policies for an agent on a toroidal $11 \times 11$ grid, with reward peaking at its center. The POMDP only observes one coordinate, whereas the MDP observes both.

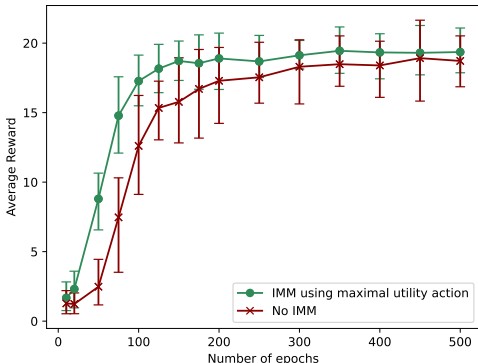

*Figure 2: Average reward of MDP trained without and with IMM incorporating POMDP. Details in Appendix D.*

## 8   Conclusion

In this paper, we addressed the question of how to incorporate accurate restricted models in the training of full-featured models using the principle of induced model matching (IMM). This was inspired by interpreting some noising-based data augmentation techniques in natural language noising, as an attempt to harness the knowledge of the restricted model used for noising. We showed that naïve noising is not the best way to incorporate this knowledge, as it may fail to consistently recover the true model. IMM, on the other hand, directly aligns the learned model with the target model and is consistent. The results shows that IMM always outperforms noising, and improvements even decay gracefully with lower restricted model quality (see Appendix D.4). One limitation of our approach is that computing the induced model exactly is not always viable. To remedy this, we proposed sampled IMM, which yields accurate but somewhat computationally demanding learning, and serialized IMM, which is slightly less accurate but has a potential to be as efficient as the no-IMM baseline. We then experimentally demonstrated the gains that IMM can offer, in a logistic regression toy example, when training LSTM language models and fine-tuning pretrained transformers, and in a simple reinforcement learning scenario. We believe that scaling from feature-restricted to full-featured models is an important yet under-studied sub-problem of knowledge transfer. In addition to the proof-of-concept experiments in this paper, many others of particular contemporary relevance may be devised. For example, lengthening the context of large language models remains an open problem; we believe that IMM can be part of betters solutions, by correctly informing new longer-context LLMs using current shorter-context LLMs. The principle behind IMM is applicable very generally and we hope this work gives impetus to such explorations.

## Acknowledgments and Disclosure of Funding

This paper is based upon work supported in part by the National Science Foundation, through the NSF CAREER Program under Award No. CCF-2146334 (From Rare Events to Competitive Learning Algorithms), the NSF HDR TRIPODS Phase II Program under Award No. ECCS-2217023 (IDEAL Institute), and the NSF TRIPODS Phase I Program under Award No. CCF-1934915 (UIC Foundations of Data Science Institute). Computational infrastructure was supported in part by the NSF MRI Program Award No. CNS-1828265 (COMPaaS DLV).

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

# A Explanation of main notations

Figure 3 gives a schematic overview of the process of IMM.

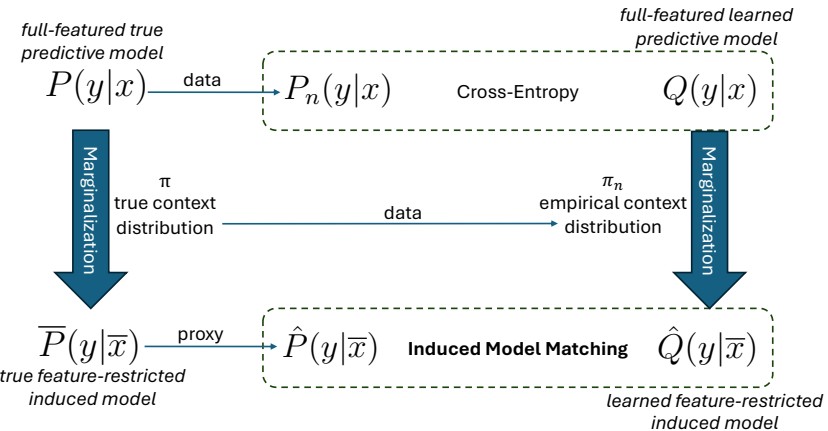

*Figure 3: Schematic overview of IMM.*

*Table 3: Glossary*

| | |
|---|---|
| Prediction Variables | $y$ (generic)    $y_t$ (at datapoint $t$ of $n$) |
| Context Variables | $x$ (generic)    $x_t$ (at datapoint $t$ of $n$) 
 $\overline{x}$ denotes short context and 
 $\underline{x}$ denotes extended context 
 Thus, $x = (\overline{x}, \underline{x})$ 
 extend($\overline{x}$) multiset of $\underline{x}$ co-occurring with $\overline{x}$ |
| Prediction based on Full Context | $P(y\|x)$ (using true model)    $Q(y\|x)$ (using learned model) 
 In the case of BERT experiments, $Q(y\|x)$ denotes a prediction 
 vector for a certain masked position. |
| Dataset Size | $n$ is the dataset size 
 The $n$ subscript may be used to denote empirical distributions |
| Context Distribution | $\pi$ (true context distribution) 
 $\pi_n$ (empirical context distribution) |
| Learned Model | $Q$ alone, without parenthesis denotes the learned context-conditional model (which 
 includes its parameters), e.g., LSTM or Transformer |
| Prediction based on Restricted Context | $\overline{P}(y\|\overline{x})$, using true induced model 
 $\hat{P}(y\|\overline{x})$, using target induced model, a proxy to $\overline{P}$. This is 
 the Kneser–Ney bigram in our experiments. 
 $\overline{Q}(y\|\overline{x})$, using learned induced model, i.e., based on true $\pi$ 
 $\hat{Q}(y\|\overline{x})$, using empirical learned induced model, i.e. based 
 on empirical $\pi_n$, a proxy to $\overline{Q}$ |

Table 3 compiles our main notation. We summarize the main formulation here and explain how it connects to its instances in the experiments.

- $Q$ is *always* the full model (LSTM or Transformer). $Q(y_t|x_t)$ is thus a *learned context-conditional model*: takes the full context ($x_t$) and outputs prediction probabilities (over $y_t$). In Eq. (4), $-\log(Q)$ is the log-loss, whose average is the cross-entropy.
  - LSTM/PTB: context = previous words, prediction = next word, and cross-entropy = main (next-word prediction) loss.

- BERT/GLUE: context = unmasked words, prediction = masked word, and cross-entropy = MLM (masked word prediction) portion of the fine-tuning loss.
  - *Note: KL divergence and cross-entropy are interchangeable, as their difference doesn't depend on Q.*

- **True context distribution** $\pi$ and **true context-conditional model** $P$ describe the unknown probability space. We assume tokenized data $(x_t, y_t)$ generated according to $\pi(x_t)$ and $P(y_t|x_t)$.
  - *The goal of learning Q is to approximate P.*

- **Empirical context distribution** $\pi_n$ and **empirical context-conditional model** $P_n$ are *histograms* of the context and context-conditional prediction, per training counts.
  - Using these instead of the true is equivalent to replacing expectations (over the data distribution) with sums (over the training set), e.g., Eq. (5) $\rightarrow$ Eq. (9).

- **Induced model**: specializes full context-conditional model $(y_t|x_t)$ to short context only $(y_t|\overline{x}_t)$, under a specific context distribution.
  - If the full context-conditional is $P$ and the context distribution is $\pi$, we get the **true induced model** $\overline{P}$. This is the ideal, or Bayes' optimal small model (which is not available in practice). In practice, we approximate it by a **target induced model**, $\hat{P}$ (e.g., Kneser–Ney bigram).
  - If the full context-conditional is $Q$, and the context distribution is...
    * ... $\pi$, we get the **learned induced model** $\overline{Q}$. This is the best way $Q$ can predict based only on the short context. However, we cannot evaluate it without $\pi$. Instead, we use...
    * ... $\pi_n$, and get the **empirical learned induced model** $\hat{Q}$. This, we *can* evaluate and use in our training.

**In summary**, the experiments use $\hat{Q}$ given by Eq. (6), efficiently approximated via sampling in Eq. (18). This is then plugged into the empirical IMM objective in Eq. (9), which pits $\hat{Q}$ against $\hat{P}$.

# B  Caveats of prior methods: noising and reverse-KD

## B.1  A quick review and analysis of noising-based data augmentation

In Xie et al. (2017), noising was proposed as an approach to perform data augmentation in language and sequence-to-sequence models. Two types of noising were suggested: context noising and target noising. We first show that context noising is not justifiable except in select cases. To simplify, we consider only a bigram model, with counts $c(x, y)$, where $x$ is the context and $y$ is the prediction. The general noising scheme in Xie et al. (2017) would, with some probability $\gamma(x)$ that may depend on the context, substitute either the context, the prediction, or both with a sample from a distribution $q(\cdot)$. It is straightforward to show that, as a result, the expected counts presented to the learner are no longer $c(x, y)$, but rather change to $\tilde{c}(x, y)$ as follows.

(a) If only contexts are noised, then $\tilde{c}(x, y) =$

$$[1 - \gamma(x)]c(x, y) + q(x) \sum_{x'} \gamma(x')c(x', y)$$

(b) If only the predictions are noised, then $\tilde{c}(x, y) =$

$$[1 - \gamma(x)]c(x, y) + q(y)\gamma(x)c(x)$$

(c) Lastly if both predictions and targets are noised, independently, then $\tilde{c}(x, y) =$

$$[1 - \gamma(x)]c(x, y) + q(y)q(x) \sum_{x', y'} \gamma(x')c(x', y')$$

These noising schemes are primarily motivated (see Xie et al. (2017), Section 3.3) through the idea that noising leads to classical forms of smoothing, and which in turn may be a desirable property

to inject into training more sophisticated language models. This is indeed true in case (a) when $\gamma(x') = \lambda$ is constant and $q(\cdot) = c(\cdot)/n$ is the unigram distribution, leading to simple interpolative smoothing. It immediately fails to be true when $q$ is any other distribution even if $\gamma(x') = \lambda$, as one needs to normalize with $c(x)$ to get a conditional distribution. The failure of this interpretation is even more apparent in case (a), when $\gamma(x')$ and $q(\cdot)$ are determined via the missing mass and Kneser–Ney backoff, failing to recreate any likeness of the Kneser–Ney smoothed bigram. The situation is at least as bad in case (c).

The only instance that succeeds to uphold "noising as smoothing" is case (b), which recreates both interpolative smoothing in the unigram case, and gives an output related to the Kneser–Ney bigram with the choices of $\gamma(x')$ and $q(\cdot)$ from the paper. This last case is of particular interest to us, namely because even though it may appear that we recreate the Kneser–Ney bigram itself, the choice of $\gamma(x') = \gamma_0 N_{1+}(x_{-1}, \cdot)/c(x)$ with $\gamma_0 = 0.2$ or $0.6$ (see Xie et al. (2017), Table 1 and Figures 1 and 2) makes it evident that this is under-weighed to represent the missing mass, which typically corresponds to larger discounts (0.75 or higher), due to the heavy Zipfian nature of language. What can we deduce from this? If $\nu(x)$ is the true missing mass in the Kneser–Ney model, then we can understand this choice as $\gamma(x) = \lambda\nu(x)$. As a result, we have:

$$\tilde{c}(x,y) = (1 - \lambda)c(x,y) + \lambda\left[(1 - \nu(x))c(x,y) + \nu(x)q(y)c(x)\right] \tag{16}$$

Upon close examination, we identify this as an interpolation between the data on the left and the Kneser–Ney bigram on the right, which suggests a form that is similar to IMM, in that the typical calculation of the log-loss is additionally joined by the log-loss pitted against a target model, the Kneser–Ney bigram.

## B.2   Proof of Proposition 6.1

Assume that we are optimizing the idealized noising objective of Eq. (14) — i.e., we are operating in the infinite-data regime — and let $Q^\star$ be its global minimizer. Assume further that the model class for $Q$ contains the true model $P$ — i.e., we are in the realizable case. Then, there exists a choice of $\pi$ and $P$ such that $Q^\star \neq P$.

*Proof.* Let us first rewrite (14) below, with the addition of constants (an entropy to the first term and $P(y|x)$ in the logarithm of te second term) that do not change the minimizer of this objective:

$$\underbrace{\mathsf{D}(P\|Q)}_{f(Q)} + \lambda \underbrace{\sum_x \pi(x) \sum_y \overline{P}(y|\overline{x}) \log \frac{P(y|x)}{Q(y|x)}}_{g(Q)}.$$

To simplify the notation, let $f$ refer to main objective term and $g$ refer to the noising regularization, which we have equivalently identified with the single-sample approximation of IMM.

We note that both $f$ and $g$ are convex in $Q$ and $f \geq 0$. Since $f(Q)$ is minimized at $P$, its gradient should be 0 at $P$. $g(Q)$ is 0 at $P$.

If $\lambda g(Q^\dagger)$ is $-\epsilon < 0$ at *some* $Q^\dagger$, then along the line connecting $P$ to $Q^\dagger$, $\lambda g(Q)$ is below the line 0 to $-\epsilon$. To compensate, $f$ would need to be *above* the line 0 to $+\epsilon$, which would violate the fact that the gradient of $f$ is 0 at $P$.

We now numerically show that $g(Q)$ can indeed be negative at *some* $Q^\dagger$, for a given construction with specific choices of $\pi$ and $P$.

Consider a trigram scenario, with $x = (x_{-1}, x_{-2})$ and where we understand the short context as $\overline{x} = x_{-1}$ and the extended context as $\underline{x} = x_{-2}$. Let us rewrite the regularization term $g(Q)$ explicitly splitting the short and long contexts:

$$g(Q) = \sum_{x_{-1}, x_{-2}} \pi(x_{-1}, x_{-2}) \sum_y \overline{P}(y|x_{-1}) \log \frac{P(y|x_{-1}, x_{-2})}{Q(y|x_{-1}, x_{-2})} \tag{17}$$

We search for distributions over the prediction $y$, short context $\overline{x}$, and extended context $\underline{x}$. Consider the following resulting tensor:

$$P(y, x_{-1}, x_{-2}) = \begin{bmatrix} \begin{bmatrix} 0.396 & 0.003 \\ 0.1 & 0.05 \end{bmatrix} \\ \\ \begin{bmatrix} 0.004 & 0.297 \\ 0.1 & 0.05 \end{bmatrix} \end{bmatrix}$$

Based on this, we get the context distribution $\pi(\overline{x}, \underline{x})$, the conditional $P(y \mid x_{-1}, x_{-2})$, as well as the restricted model $\overline{P}$

$$\pi(x_{-1}, x_{-2}) = \begin{bmatrix} 0.4 & 0.3 \\ 0.2 & 0.1 \end{bmatrix}$$

$$P(y \mid x_{-1}, x_{-2}) = \begin{bmatrix} \begin{bmatrix} 0.99 & 0.01 \\ 0.5 & 0.5 \end{bmatrix} \\ \\ \begin{bmatrix} 0.01 & 0.99 \\ 0.5 & 0.5 \end{bmatrix} \end{bmatrix}$$

These give us the following induced model:

$$\overline{P}(y \mid x_{-1}) = \begin{bmatrix} 0.57 & 0.5 \\ 0.43 & 0.5 \end{bmatrix}$$

Now consider the following choice of $Q^{\dagger}$:

$$Q^{\dagger}(y \mid x_{-1}, x_{-2}) = \begin{bmatrix} \begin{bmatrix} 0.5 & 0.5 \\ 0.5 & 0.5 \end{bmatrix} \\ \\ \begin{bmatrix} 0.5 & 0.5 \\ 0.5 & 0.5 \end{bmatrix} \end{bmatrix}$$

Plugging the above choice of $\pi$, $\overline{P}$, $P(y \mid x_{-1}, x_{-2})$ and $Q$ into Eq. (17) gives us a negative value of $g(Q^{\dagger}) = -1.1$, thus completing the construction of the counterexample and the proof. □

Table 4: *Deterioration of noising vs. consistency of IMM when $\lambda = 1.5$ is fixed in the logistic regression example. This is a concrete manifestation of Propositon 6.1.*

| Dataset Size | Baseline | Noising | IMM | IMM-Noising Gap |
|---|---|---|---|---|
| N=5 | 73.14 +16.17/-14.53 | 76.85 +18.58/-13.15 | 79.96 +14.99/-12.38 | 3.11 |
| N=10 | 84.17 +13.51/-10.86 | 84.37 +7.40/-7.63 | 88.65 +9.65/-7.68 | 4.28 |
| N=15 | 89.86 +8.86/-6.80 | 86.17 +6.17/-5.53 | 92.52 +6.19/-4.81 | 6.35 |
| N=20 | 92.35 +7.35/-5.32 | 86.99 +5.69/-5.37 | 94.30 +4.63/-3.70 | 7.30 |
| N=30 | 94.94 +3.97/-3.43 | 88.68 +4.35/-4.32 | 95.69 +3.39/-2.64 | 7.01 |
| N=40 | 96.33 +3.00/-2.33 | 89.70 +3.73/-3.97 | 96.59 +2.59/-2.08 | 6.89 |
| N=50 | 97.14 +2.47/-2.20 | 90.78 +3.78/-3.55 | 97.34 +2.34/-1.99 | 6.56 |

**Experimental Evidence of Inconsistency** One may dismiss Proposition 6.1 as being too specific of a counterexample. After all, in Figure 1, the gap between all methods does vanish. It is important therefore to emphasize that this only happens because we are being very favorable to noising. Specifically, we are decaying $\lambda$ (the amount of noising) optimally with increasing data. This is *necessary* for noising, for the precise reason of Proposition 6.1: even if the target model is perfect, because noising incorrectly tracks the target model, without decaying its influence it will not only not narrow the gap, but would in fact derail the learned model. Decaying $\lambda$ is also acknowledged as

critical in the reverse knowledge-distillation literature (see for example Sec. 3 of Qin et al. (2021)). However, tuning $\lambda$ is *optional* for IMM, thanks to $\hat{Q}$ (with more extended context samples) accurately tracking the target (see the flat curves on the right column of Figure 6 in Appendix D.1.1).

To experimentally verify this, we ran the logistic regression example with fixed $\lambda = 1.5$ (optimal at data size 5). The results are below. IMM maintains performance comparable to Figure 1, whereas noising experiences a widening gap, and soon underperforms even the baseline.

### B.3 Knowledge Distillation Theory

We would like mention that recent theory elucidates how weak teachers can be useful in reverse KD (Kaplun et al., 2022). That work is specific to the classification setting and assumes that the teachers are good samplers, i.e., noisy versions of the Bayes decision, and shows that reverse-KD can effectively ensemble multiple teachers and remove the noise. This setting and assumptions do not directly apply here, however, a common insight may be that the process of inducing a model can also be thought of as ensembling, but over contexts rather than over independent resampling. Otherwise, considering that the implicit noising objective that we identify in Eq. (17) is equivalent to the explicit objective of reverse knowledge distillation, the arguments in this section equally apply as caveats for reverse-KD when the weak teacher is a restricted-context model.

## C    Implementation Details

---

**Algorithm 1** Sampled IMM with SGD for a Model $Q$ with parameters $W$. To connect with the notation of the paper, this version most closely resembles the language modeling version (with the details of the sequentialization of log-sum gradients skipped for simplicity). The logistic regression equivalent uses a soft nearest neighbor density estimate (ref D.1) instead of creating a multiset and sampling from it.

---

**Input:** Data $(x_t, y_t)$ for $t = 1, 2, ..., n$, $k$ = sampling rate
**Output:** IMM-trained model $Q$
1: **repeat**
2:     $Q(y|x_t) \leftarrow \text{FEEDFORWARD}(Q, \overline{x}_t, \underline{x}_t)$
3:     $\nabla_W \textsf{Cross-Entropy}(Q) \leftarrow \text{BACKPROPAGATE}(Q, \textsf{Cross-Entropy}(Q, y_t))$
4:     $\hat{Q}(y|\overline{x}_t) \leftarrow 0$
5:     **for all** $i, \underline{x}^{\textsf{sample}} \in \textbf{Enumerate}(\textbf{Sample}(\text{extend}(\overline{x}_t), k))$ **do**
6:         $\hat{Q}(y|\overline{x}_t) \leftarrow \hat{Q}(y|\overline{x}_t) + \text{FEEDFORWARD}(Q, \overline{x}_t, \underline{x}^{\textsf{sample}})$
7:     **end for**
8:     $\hat{Q}(y|\overline{x}_t) \leftarrow \hat{Q}(y|\overline{x}_t)/k$
9:     $\textsf{IMM}_t(Q) = -\sum_y \hat{P}(y|\overline{x}_t) \log \hat{Q}(y|\overline{x}_t)$
10:     $\nabla_W \textsf{IMM}_t(Q) \leftarrow \text{BACKPROPAGATE}(Q, \textsf{IMM}_t(Q))$
11:     $\text{APPLYGRADIENTS}(\nabla_W \textsf{Cross-Entropy}(Q) + \lambda \nabla_W \textsf{IMM}_t(Q))$
12: **until** convergence

---

### C.1    k-approximated IMM

For the approximation, a given short context $\overline{x}_t$, we take only a fixed number $k$ of samples $X_1, \cdots, X_k$ uniformly from extend($\overline{x}_t$). The approximated $\hat{Q}$ (Eq. (11)) can then be written as

$$\hat{Q}(y|\overline{x}) \approx \frac{1}{k} \sum_{i=1}^{k} Q(y|\overline{x}_t, \underline{x}_i^{\textsf{sample}} \sim \text{extend}(\overline{x}_t)) \tag{18}$$

Indeed this is what is happening on lines 5-8 of Algorithm 1. Sampling $k$ extended contexts is only part of the solution. Another algorithmic innovation that we need is to address the task of computing the derivative of our main objective, Eq. (8), because it requires differentiating Eq. (7), where a naïve implementation would need all $k$ instances of the LLM used to estimate $\hat{Q}(y|\overline{x})$ in the memory simultaneously. Fortunately, we provide a solution to this, explain in detail in Section C.2 below.

**Algorithm 2** Serialized IMM with SGD for a Model $Q$ with parameters $W$. As in Algorithm 1, this is notationally closest to the language modeling version. For implementation details for the logistic regression example, see D.5, as well as D.1 (for the soft nearest-neighbor density estimate formulation).

---

**Input:** Data $(x_t, y_t)$ for $t = 1, 2, ..., n$, $r =$ refresh frequency
**Output:** IMM-trained model $Q$

1: **repeat**
2:     $Q(y|x_t) \leftarrow \text{FEEDFORWARD}(Q, \overline{x}_t, \underline{x}_t)$
3:     $\nabla_W \text{Cross-Entropy}(Q) \leftarrow \text{BACKPROPAGATE}(Q, \text{Cross-Entropy}(Q, y_t))$
4:     **if** Repeats $\%$ $r = 0$ **then**
5:         $Q_\dagger \leftarrow Q$
6:         $\hat{Q}_\dagger \leftarrow 0$
7:         **for all** $t = 1, 2, ..., n$ **do**
8:             **for all** $y$ **do**
9:                 $\hat{Q}_\dagger(y|\overline{x}_t) \leftarrow \hat{Q}_\dagger(y|\overline{x}_t) + \text{FEEDFORWARD}(Q_\dagger, x_t)$
                {Or use other induction mechanism, e.g., density estimator for logistic regression.}
10:             **end for**
11:         **end for**
12:         $\text{NOGRADIENT}(Q_\dagger, \hat{Q}_\dagger)$
13:     **end if**
14:     $\widetilde{\text{IMM}}_t(Q) := - \sum_y \frac{Q_\dagger(y|x_t)}{\hat{Q}_\dagger(y|\overline{x}_t)} \hat{P}(y|\overline{x}_t) \log Q(y|\overline{x}_t)$
    {This is *not* the IMM loss, but its gradient averages to the correct gradient, see Section 5.}
15:     $\nabla_W \widetilde{\text{IMM}}_t(Q) \leftarrow \text{BACKPROPAGATE}(Q, \widetilde{\text{IMM}}_t(Q))$
16:     $\text{APPLYGRADIENTS}(\nabla_W \text{Cross-Entropy}(Q) + \lambda \nabla_W \widetilde{\text{IMM}}_t(Q))$
17: **until** convergence

---

### C.2 IMM Gradient Computation

Eq. (7) is a cross-entropy and the $\hat{Q}(y|\overline{x})$ term that we approximated in Eq. (18) occurs inside the $\log$ term of the cross-entropy. Naïvely backpropagating through this term makes the memory complexity of backpropagation scale with the number of random samples $k$ used to approximate $\hat{Q}(y|\overline{x})$. In our experiments, it was problematic for the GPU (Nvidia V100 with 32 GB memory) to perform backpropagation for $k > 6$ for the LSTM RNN.

Contemporary deep learning frameworks do not have the ability to sequentialize along $k$ the computation of the derivative of a $\log$ where the argument of the $\log$ is an average (or a sum) of $k$ entities. This is expected, because $\log(A + B)$ cannot be decomposed, and therefore, sequentializing this by splitting the cost function (this cross-entropy term) is not possible.

Despite this, we propose a solution that may be of interest generally when such differentiation needs to be performed. This stems from the simple observation that the derivative of $\log(f + g)$ can be represented as a weighted average of the derivatives of $\log(f)$ and $\log(g)$, where the "crosstalk" weights do not require differentiation:

$$\nabla \log(f + g) = \frac{\nabla f + \nabla g}{f + g} = \tfrac{f}{f+g} \nabla \log f + \tfrac{g}{f+g} \nabla \log g.$$

In summary, this allows us to compute the derivatives separately for each random sample and accumulate them appropriately.

### Sequentializing IMM gradient computations using crosstalk

Recall that we can write an IMM component at data point $t$ (Eq. (12)) with $k$ randomly sampled extended contexts as:

$$\text{IMM}_t(Q) = \text{Cross-Entropy}(\hat{Q}(y|\overline{x}), \hat{P}(y|\overline{x}))$$

$$= -\sum_y \hat{P}(y|\overline{x}) \log\left[\mathbf{E}_{\underline{X}}\left[Q(y|\overline{x}_t, \underline{X}) \mid \overline{x}_t\right]\right]$$

$$\overset{\star}{\approx} -\sum_y \hat{P}(y|\overline{x}) \log\left[\frac{1}{k}\sum_{i=1}^{k} Q(y|\overline{x}_t, \underline{X}_i \sim \text{extend}(\overline{x}_t))\right] \qquad (19)$$

$$= -\sum_y \hat{P}(y|\overline{x}) \log\left[\frac{1}{k}\sum_{i=1}^{k} Q(y|\overline{x}_t, \underline{x}_i)\right]$$

where $\star$ comes from Eq. (18).

The above entity cannot be directly decomposed into $k$ terms because it involves the log of a sum. Since it's the IMM component of the loss, during backpropagation, we will need its derivative with respect to the parameters of the neural model (LSTM RNN or BERT). In the next section, we see that the derivative can be decomposed into $k$ terms.

Denoting the set of parameters using $W$, we can then write $\nabla_W \text{IMM}(Q)$ as below.

$$\nabla_W \text{IMM}_t(Q) = -\sum_y \hat{P}(y|\overline{x}) \nabla_W \log\left[\frac{1}{k}\sum_{i=1}^{k} Q(y|\overline{x}_t, \underline{x}_i)\right]$$

$$= -\sum_y \hat{P}(y|\overline{x}) \frac{\nabla_W\left[\frac{1}{k}\sum_{i=1}^{k} Q(y|\overline{x}_t, \underline{x}_i)\right]}{\frac{1}{k}\sum_{i=1}^{k} Q(y|\overline{x}_t, \underline{x}_i)}$$

$$= -\sum_y \hat{P}(y|\overline{x}) \frac{\frac{1}{k}\sum_{i=1}^{k} \nabla_W Q(y|\overline{x}_t, \underline{x}_i)}{\frac{1}{k}\sum_{i=1}^{k} Q(y|\overline{x}_t, \underline{x}_i)}$$

$$= -\sum_y \hat{P}(y|\overline{x}) \frac{\sum_{i=1}^{k} \nabla_W Q(y|\overline{x}_t, \underline{x}_i)}{\sum_{i=1}^{k} Q(y|\overline{x}_t, \underline{x}_i)}$$

$$= -\sum_y \hat{P}(y|\overline{x}) \left[\sum_{i=1}^{k}\left(\frac{1}{\sum_{i=1}^{k} Q(y|\overline{x}_t, \underline{x}_i)}\right) \nabla_W Q(y|\overline{x}_t, \underline{x}_i)\right] \qquad (20)$$

$$= -\sum_y \hat{P}(y|\overline{x}) \left[\sum_{i=1}^{k}\left(\frac{Q(y|\overline{x}_t, \underline{x}_i)}{\sum_{i=1}^{k} Q(y|\overline{x}_t, \underline{x}_i)}\right) \frac{\nabla_W Q(y|\overline{x}_t, \underline{x}_i)}{Q(y|\overline{x}_t, \underline{x}_i)}\right]$$

$$= -\sum_y \hat{P}(y|\overline{x}) \left[\sum_{i=1}^{k}\left(\frac{Q(y|\overline{x}_t, \underline{x}_i)}{\sum_{i=1}^{k} Q(y|\overline{x}_t, \underline{x}_i)}\right) \nabla_W \log Q(y|\overline{x}_t, \underline{x}_i)\right]$$

$$= -\sum_{i=1}^{k}\sum_y \hat{P}(y|\overline{x}) \left(\frac{Q(y|\overline{x}_t, \underline{x}_i)}{\sum_{i=1}^{k} Q(y|\overline{x}_t, \underline{x}_i)}\right) \nabla_W \log Q(y|\overline{x}_t, \underline{x}_i)$$

$$= \sum_{i=1}^{k} \nabla_W \left[-\sum_y \hat{P}(y|\overline{x}) \underbrace{C_{t,i}(y)}_{\text{crosstalk}} \log Q(y|\overline{x}_t, \underline{x}_i)\right]$$

We notationally move the derivative outside, but it is crucial that $C_{t,i}(y) = \frac{Q(y|\overline{x}_t, \underline{x}_i)}{\sum_{i=1}^{k} Q(y|\overline{x}_t, \underline{x}_i)}$ be treated as a constant with respect to the parameters of the LLM. We do this to show that we can effectively compute individual cross-entropies, with the $C_{t,i}(y)$ terms acting like a "crosstalk" between the $k$ random samples. This exactly parallels the decomposition that we highlighted above for the case of $\nabla \log(f + g)$. Note that this decomposition is not possible directly on the cross-entropy (that involves the log of a sum), but possible when we consider its gradient.

**Overcoming limitations of contemporary frameworks to compute the crosstalk vector**

We cannot compute the above total gradient, i.e. $\nabla_W \mathsf{IMM}(Q)$ unless for every $t$ and $y$ we have $C_{t,i}(y)$ for $i \in [1, k]$. To this end, we implement this by partially running the feedforward computation graphs of all $k$ random samples only until the point which gives us $Q(y|\overline{x}_t, \underline{x}_i)$ for $i \in [1, k]$. Once we have all the $Q(y|\overline{x}_t, x_i)$, we can normalize all of them by their element-wise sum to get the weights $C_{t,i}(y)$. These weights are then fed back into the graph, and the forward pass is completed to get a "tampered" cross-entropy for $i \in [1, k]$. The $k$ backward passes using these tampered cross-entropies give us the terms inside the sum in the last step of (20), which after summation produce $\nabla_W \mathsf{IMM}(Q)$.

# D  Additional Experimental Details

We now provide some additional detail of both the logistic regression and language modeling experiments. One minor difference between the two is that in the logistic regression case we interpolate the main loss $(1 - \lambda)$ and the IMM risk $(\lambda)$, to efficiently cross-validate for the choice of $\lambda$ as we explain in Section D.1.1. In the language modeling experiments, we simply add the regularizer with the factor $\lambda$, while maintaining the main loss as is (factor 1).

## D.1  Logistic Regression Toy Example

We provide the complete details of the logistic regression toy example[2]. Consider a dataset with feature space $x = (x_1, x_2, x_3) \in \mathbb{R}^3$ containing two linearly separable classes, defined by a linear discriminant $g(x) = ax_1 + bx_2 + cx_3 + d$. We will think of features as decomposable into $\overline{x}$, the *short context* of restricted features, and $\underline{x}$, the *extended context* of remaining features. For this example, let $\overline{x} = x_1$ and $\underline{x} = (x_2, x_3)$. The choice of word "context" stems from features in language modeling.

**Target restricted model**  Say features are sampled uniformly over a cube and that we have ample observation of the labels $y = \mathbf{1}\{g(x) > 0\}$ along only the short context, i.e., ample data points of the form $(x_1, y)$. This allows us to build an excellent *restricted model* to predict the label based on just $x_1$, call it $\overline{P}(y|x_1)$ or $\overline{P}(y|\overline{x})$. For the sake of this example, let us assume that this is the (restricted) Bayes predictor or true conditional probability.

**Main question**  We then observe a few samples of all features along with labels, i.e., data points of the form $(x_1, x_2, x_3, y)$. How can we use the restricted model $\hat{P}(y|x_1)$ as part of the training of the *full model* $Q(y|x_1, x_2, x_3)$?

**Induced model**  Say we have $n$ data points. Let us index them by $t = 1, \cdots, n$, written either as $(x_t, y_t)$ or $(x_{1,t}, x_{2,t}, x_{3,t}, y_t)$. The key concept of the method is that **it is not $Q$'s behavior that should *target* that of $\hat{P}(y|x_1)$, but rather the behavior of $Q$ if it were itself restricted**. We call this restricted version $\overline{Q}$ the *induced model* of $Q$ and, by marginalization, we interpret it as the average of $Q$'s predictions, when $\underline{x}_t$ is drawn from its conditional distribution given $\overline{x}_t$. Since we typically don't have access to this distribution, we approximate it empirically. In language modeling, we could just sample from the empirical distribution of $\underline{x}$ for a given $\overline{x}$. In logistic regression, this is not viable since $x_1$ does not repeat. We instead use a soft nearest-neighbor density estimate $\hat{f}(x_2, x_3|x_1) \propto \sum_{t=1}^{n} \delta_{x_{2,t}, x_{3,t}}(x_2, x_3) e^{-\alpha|x_{1,t} - x_1|}$, where $1/\alpha$ is the bandwidth of the kernel. (With cross-validation, we determine $\alpha = 1$ to be a good choice in this example.) If we let $w_t = e^{-\alpha|x_{1,t} - x_1|}$, the resulting induced model by marginalization is:

$$\overline{Q}(y|\overline{x}) = \int f(\underline{x}|\overline{x}) Q(y|\overline{x}, \underline{x}) \tag{21}$$

$$\approx \sum_{t=1}^{n} \frac{w_t(x)}{\sum_{t=1}^{n} w_t(x)} Q(y|\overline{x}, \underline{x}_t) \tag{22}$$

Just like in language modeling, to induce a model we need to to draw $\underline{x}$'s from its conditional distribution given $\overline{x}$. Unlike in the discrete case where we could directly approximate the conditional distribution using the empirical counts, we need to rely here on an estimated density.

---

[2]Code is available at https://github.com/uicdice/imm-logistic-regression

**Matching the target** The main objective of logistic regression is to minimize $-\sum_{t=1}^{n} \log Q(y_t|x_{1,t}, x_{2,t}, x_{3,t})$, which is equivalent to Cross-Entropy$(Q)$ versus the empirical distribution. We now additionally want the induced model $\overline{Q}$ to *match* the target restricted model $\hat{P}$. This gives this method the name Induced Model Matching (IMM). This requirement can be captured through a KL-divergence, and introduced as a regularizer. The result is a secondary objective, which we call the *IMM risk*, expressed as:

$$\mathsf{IMM}(Q) = \sum_{t=1}^{n} \sum_{y=0,1} \hat{P}(y|x_{1,t}) \log \frac{1}{\overline{Q}(y|x_{1,t})} \tag{23}$$

The overall objective then becomes

$$\mathsf{Cross\text{-}Entropy}(Q) + \lambda\,\mathsf{IMM}(Q), \tag{24}$$

where $\lambda$ is the regularization trade-off and can be determined via cross-validation (details in the Appendix D.1.1).

**IMM improves restricted-context prediction** In Figure 4, for $n = 10$ and 30 Monte-Carlo runs, we show how the induced model of $Q$ (i.e. $\overline{Q}$) fairs in its prediction, compared to the target model $\hat{P}$. Note that without IMM (i.e. with $\lambda = 0$), the secondary objective is large and thus $\overline{Q}$'s performance is worse. With increasing $\lambda$, IMM improves this performance both in average and in variance. We deduce that there is information in the accurate restricted model $\overline{P}(y|\overline{x})$ that is not known to $Q$, unless we explicitly incorporate it. This shows that $Q$ gets better at predicting from a restricted context, naturally bringing up the question: *does it also get better at predicting from the full context, i.e., the main objective?*

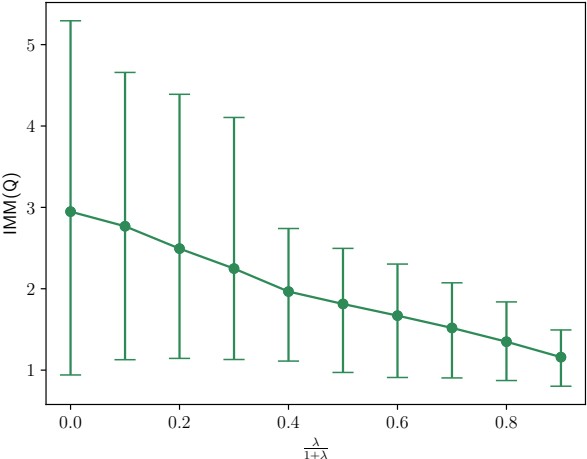

*Figure 4: Performance on restricted task, i.e. $\mathsf{IMM}(Q)$ measured on models $Q$ trained using Eq. (24) as the objective with varying $\frac{\lambda}{1+\lambda}$ ratio (refer to Appendix D.1.1). We stop at $\frac{\lambda}{1+\lambda} = 0.9$ because $\frac{\lambda}{1+\lambda} = 1$ would zero out the contribution of the main objective (and replacing the main objective completely is never the intention).*

**IMM improves full-context prediction** In Figure 1, we compare the performance of IMM-trained $Q$ (green) to that without IMM (maroon). We sweep a range of $n$ from 2 to 50, and use a cross-validation optimized $\lambda$ for each (details in the Appendix D.1.1). The key observations are that: (1) IMM always improves on the baseline performance, (2) the variance of the outcomes is also typically diminished, (3) the improvement is greater with less data, but the gain across data sizes is equivalent to access to an average of 30% extra data. This and similar experiments suggest that gains are highest when the dataset size is comparable to the number of parameters. This simple scenario demonstrates how IMM effectively harnesses the benefit of accurate feature-restricted models when training full-featured models.

**Visualizing the effect of IMM**   In Figure 5, we illustrate the 3-dimensional logistic regression problem. The features are samples uniformly in this box. The Bayes-optimal restricted model only uses the $x_1$-coordinate, and so assigns probabilities proportionally to the blue/red areas in the illustrated slice. IMM then encourages the full logistic model to be consistent with these weights, i.e., making sure the proportion of points labeled $\pm$ agrees with these weights at each $x_1$. Intuitively, this biases the separating plane to have the right inclination/alignment with the $x_1$-axis, which subsequently speeds up the learning process.

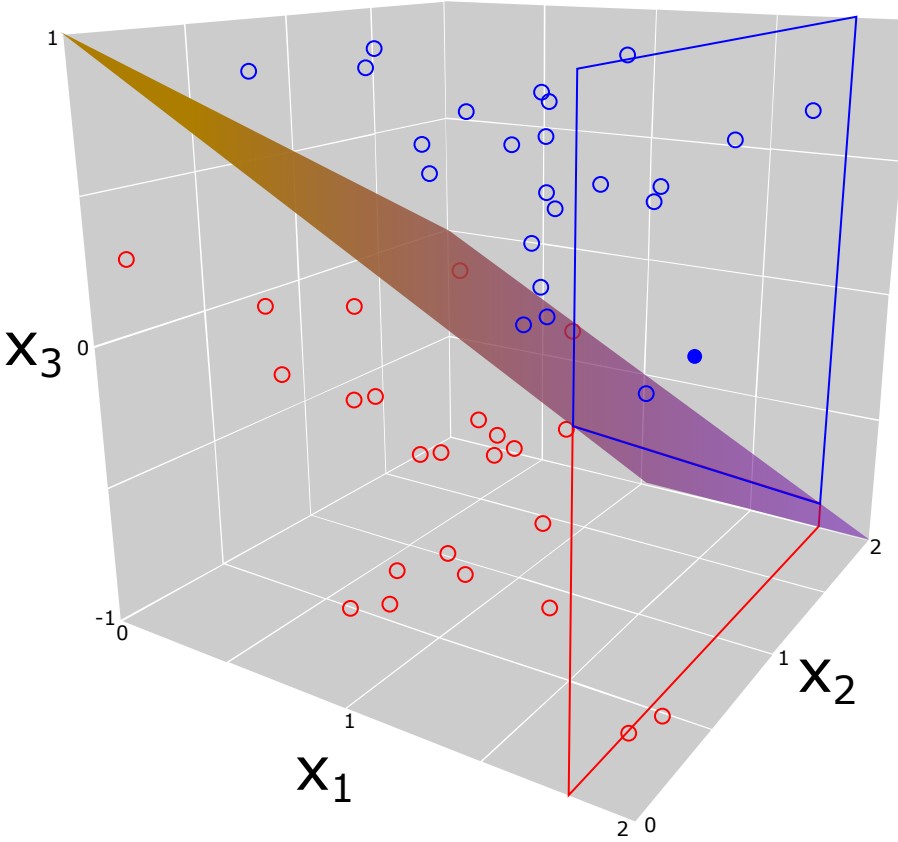

Figure 5: A visualization of the inductive bias brought upon by IMM in the logistic regression example.

### D.1.1   Tuning Lambda

As mentioned in Section 7.1, IMM always improves baseline performance. The main parameters to choose are $\alpha$ and $\lambda$. For $\alpha$, we performed a few experiments on held-out data and determined the value of $1$ to be generally adequate. For $\lambda$, we saw that adapting to the dataset size made a more significant difference. This makes sense, as for smaller datasets, we want more IMM contribution to compensate for the lack of data (i.e., we want a larger $\lambda$).

To determine a good schedule for $\lambda$ as a function of dataset size $n$, we sweep the ratio $\frac{\lambda}{1+\lambda}$ (which represents the IMM coefficient as a fraction of the combined IMM and regular loss coefficients) in the range $[0, 1]$. We repeat this for learning rates 0.1 as well as 1.0, for a range of dataset sizes, from a minimum of 2 to a maximum of 50.

We provide our $\lambda$ tuning plots in Figure 6. Since a learning rate of 1.0 was found to give the best results, we choose it for all the reported experiments. Looking at our plots in Figure 6, we select $\lambda/(1+\lambda)$ of 0.8, 0.7, 0.6 for dataset sizes, $n$ of 2, 10 and 20 respectively. These suggest a linearly decaying optimal $\lambda/(1+\lambda)$, in this range of dataset sizes. For this example, doing an interpolation fit, we create our automatic $\lambda$ schedule rule to be $\frac{\lambda}{1+\lambda} = -0.0111n + 0.818$.

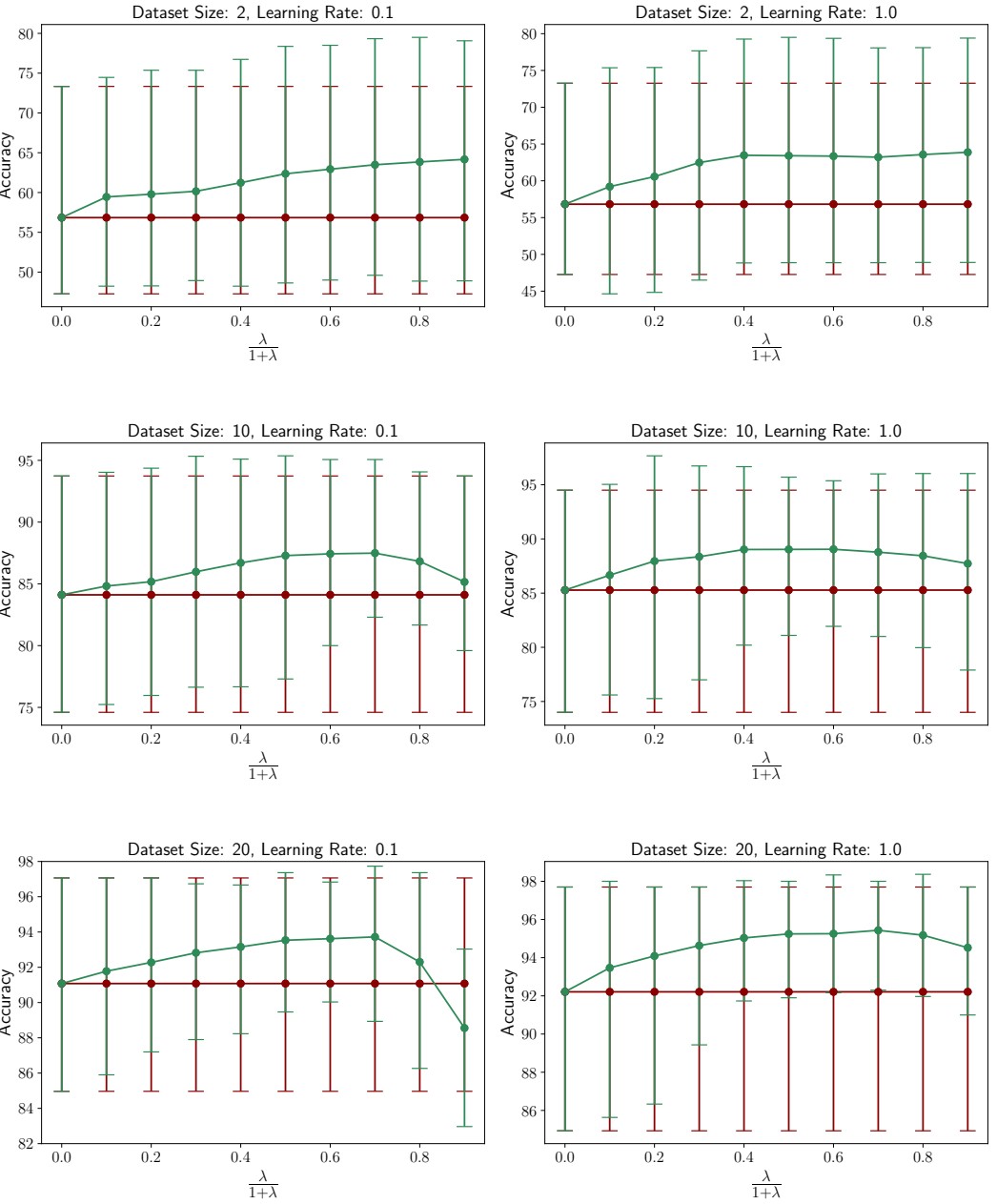

*Figure 6: Effect of varying $\lambda/(1+\lambda)$ on different dataset sizes and learning rates of $0.1$ and $1.0$. Every point is an average of 30 runs and the error bars are computed using $10^{th}$ and $90^{th}$ percentiles.*

Table 5: Restricted vs. Full Model on the Bigram Task (i.e., predicting the next word given only *the previous one.*) *The small model is the Kneser–Ney bigram on the PTB training set. The full model is the 1,500-dimensional 2-layer LSTM from Xie et al. (2017), trained without noising on the PTB training set. To make the LSTM predict the next word based only on the previous one, we complete the history by averaging predictions over all extended contexts in the training set (full word history) that share that short context (same previous word). In the language of the paper, this is exactly the* induced bigram *of the learned LSTM. We report the cross-entropy and perplexity of each model on the PTB test set. Note that the small model clearly outperforms the LSTM on this restricted task.*

| Dataset | LSTM w/o IMM | | LSTM w/ IMM | Kneser–Ney |
|---|---|---|---|---|
| Train | 260.16 | $\rightarrow$ | 237.55 | 92.19 |
| Validation | 339.24 | $\rightarrow$ | 302.30 | 199.28 |
| Test | 309.56 | $\rightarrow$ | 278.48 | 185.71 |

## D.2 Language Modeling Experiments

Code is available at https://github.com/uicdice/imm-language-modeling

### D.2.1 Evidence that IMM acts through improving restricted task

We give an illustrative example that shows the advantage that a restricted model may have, and which we would like to incorporate into the full-featured model. Clearly, our approach is only valuable if this the case. The simple experiment that we propose is to compare how well a full-featured model (LSTM) performs a restricted task, namely bigram prediction, vs. a restricted model built for that task (Kneser–Ney Bigram).

How can an LSTM perform bigram prediction when given only a single word? It can do so by averaging its output over all possible long histories consistent with that word, which is exactly the notion of induced model of Eq. (3). In Section 4, we show how to do this empirically.

When we do this comparison for the non-noised LSTM in (Xie et al., 2017), we obtain the results tabulated in Table 5. The Kneser–Ney bigram (restricted model) outperforms the LSTM (full model) on the restricted task. This may appear surprising, however, had this not been the case, the improvements that Xie et al. (2017) obtained through noising would have been hard to explain! More importantly, using IMM improves the performance of the LSTM on this task, mirroring the logistic regression case — cf. Figure 4.

### D.2.2 Parameter details

**Choice of $\lambda$** For Language Modeling experiments, due to computational cost, we did not perform an exhaustive search for the best $\lambda$. We tried a few values of $\lambda$ and tested on held-out data, and determined the value of $0.2$ to work well in most cases, and adhered to it in all of the experiments.

**Restarts** We would like to clarify that the baseline for LSTM RNN (Xie et al., 2017) reports the best perplexity across multiple restarts while the baseline for BERT (Devlin et al., 2018) reports the average across multiple restarts. We do likewise in our experiments. We have additionally included error bars for BERT experiments as that's where most of the gain is seen for BERT.

**Clipping** On the technical front, we note that in either LSTM or BERT, if gradient clipping (Pascanu et al., 2013) is used, then gradients should be clipped separately for the primary objective and the IMM objective (i.e. they should be clipped before they are added together). The reason is that they are obtained from two loss functions that have very different scales.

**Choice of $k$** To reduce overhead for LSTM, we perform IMM only every $j$'th batch using $k$-samples. Otherwise we use 1-sample IMM. For BERT, we perform $k$-sample IMM every minibatch. For the selection of the parameter $k$ (number of long histories used to approximate the induced model), we experimented with $k \in \{1, 5, 10, 20\}$. For the LSTM experiments, there was not much gain going from 10 to 20, so we settled for $k = 10$ (and $j = 5$) as a good compromise between the accuracy and time complexity of the induced model computation. For the BERT experiments, we used $k = 5$.

**IMM through reintroduced MLM loss in BERT fine-tuning**  BERT on GLUE differs from LSTM on PTB in two ways: (1) it is not a next-word predictive model, and (2) its main loss during fine-tuning is the task (e.g., CLS) loss. We do not touch the latter. Instead, we reintroduce the MLM loss, to have a fine-tuning predictive task close to our formulation. Since MLM is based on cross-entropy, IMM can be directly added to it as a secondary loss. Additionally, **only for the computation of the IMM component**, at each location of a fine-tuning training sentence, we mask that word and all future words during the forward passes being done to compute the IMM component. This forces BERT to act like a causal model, i.e., it predicts the next word based only on the past, as in our main formulation. This is done to ensure that the induced model of BERT is the right induced model to match against an accurate causal model. Incidentally, for this accurate target model we used the Kneser–Ney bigram constructed from the Wikipedia data, on which BERT is initially trained.

In summary, in the BERT experiments we fine-tune with the sum of 3 losses: the task loss, the MLM cross-entropy loss, and the IMM risk for the MLM task.

### D.3 RL Experiment

We consider an $11 \times 11$ toroidal grid in the $(x, y)$ plane with a predefined reward landscape that has a single peak at the center of the grid, see the heatmap in Figure 7. The toroidal configuration allows the agent to wrap around the grid in case it exceeds its confines, which allows us to rely on uniformity in the action space, in any state.

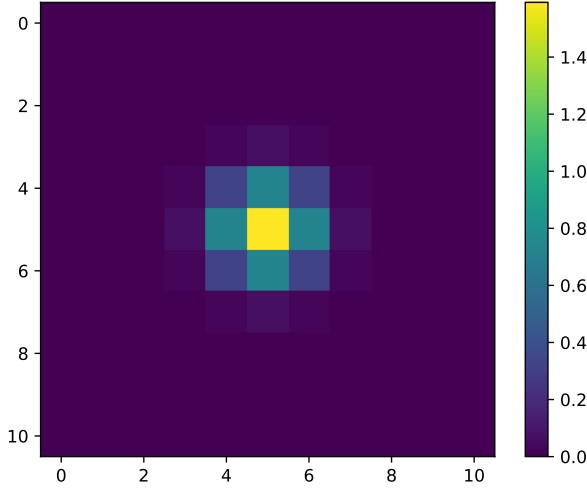

*Figure 7: Heat map representing the reward function, which depends on state only, on the $11 \times 11$ toroidal grid of the RL experiment.*

We model this as both a Markov Decision Process (MDP) as well as a Partially Observable Markov Decision Process (POMDP). The MDP agent knows both the $x$ and the $y$ coordinates of position, while the POMDP agent knows only $x$ but maintains a uniform belief over the $y$ dimension. We first solve the POMDP using the `POMDPs.jl` (Egorov et al., 2017) package package and the Fast Informed Bound (FIB) Solver (Kochenderfer et al., 2022) to obtain a very accurate policy based on partial information. (These POMDP policies are represented in the form of alpha vectors.)

**MDP Training without IMM**  Our MDP training algorithm is REINFORCE (Williams, 1992). REINFORCE, at every epoch, samples a set of observations, which can also be called a batch. In REINFORCE, at every epoch we sample fresh data from the policy. We elect to limit the observation length at every epoch, to emulate limited exploration, and hence the number of epochs can be seen as the effective dataset size.

**Adding IMM to REINFORCE**  For every observation, we consider its restricted context to be $x$ (the known dimension of the POMDP agent). Unlike language modeling experiments (where we randomly sample for the extended context), we use the POMDP belief. Here, we thus use a

uniform distribution over the unknown dimension $y$ (the analog of the extended context). Using this distribution, we compute our policy's induced model. We then match this induced model against the POMDP agent's action, which is an action based solely on the knowledge of $x$ (and only a belief over $y$). We manually set $\lambda$ to 0.25.

**Experimental Results**    We set the per epoch observation length to 50. Since then the number of epochs represents our dataset size control, we vary it and compare the performance of REINFORCE and IMM-augmented REINFORCE[3] by evaluating their average reward over a given rollout horizon. Similarly to the logistic regression experiments, we see that IMM is uniformly beneficial, with gains at their best when the dataset size is roughly in the regime of the number of parameters. We perform 30 Monte Carlo runs at each number of epochs, and report 10th and 90th percentiles. Here also, IMM reduces the variance of the reward. Results are reported in Figure 2.

### D.4    Effect of Restricted Model Quality

While the paper is presented under the assumption that we have access to a very good target model $\hat{P}$, we also ran experiments to show the effect of model quality on IMM. Of course, for the language model, we don't have a gold reference and the fact that IMM achieves gains and improves on noising is evidence that even non-perfect models help. However, for the logistic regression and RL experiments we do have gold references, which we can weaken. We weaken the restricted Bayes-optimal predictor in the regression example by adding Bernoulli noise to the label (interpolate with a coin flip), 20% for the medium quality model and 50% for the low quality model. For the RL experiment, we replace the max on the POMDP utility with a softmax, and adjust the quality via the temperature of the softmax (small is higher quality). Even with reduced model quality, we find out that IMM always improves learning. However, for this to happen, we need to tune $\lambda$ more carefully than when the model is perfect, leading to less reliance on the IMM risk when the dataset size increases (otherwise the noise in the lower quality models would offset the convergence). For logistic regression, this is illustrated in Figure 8. For RL, this is illustrated in Figure 9.

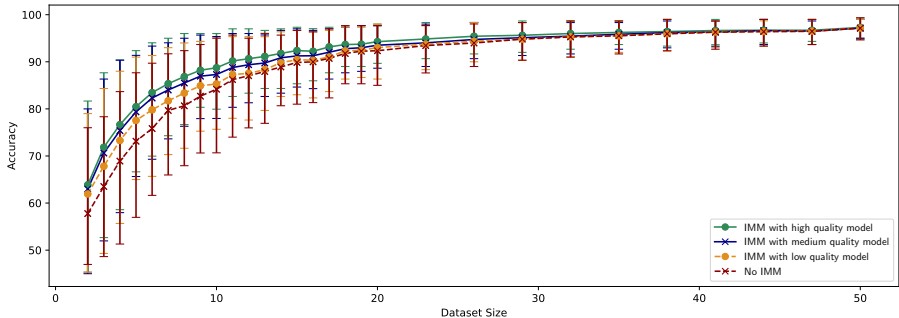

*Figure 8: Test accuracy of logistic regression model trained without IMM and with IMM using restricted model of varying quality levels, with $\lambda$ determined through dataset size (refer to Appendix D.1.1). At every dataset size, we perform 300 runs and plot the 10th and 90th percentiles for error bars.*

All experiments in this section use the Sampled IMM variant. In Appendix D.5, we show that Serialized IMM achieves competitive statistical performance with

### D.5    Serialized IMM Performance

To demonstrate the validity of the Serialized IMM variant, which comes with computational advantages that can allow IMM to scale, we implemented it for the logistic regression example. However, we first elaborate on the details of the Sampled IMM, to show the dramatic computational advantage. By combining the density estimation approach for obtaining the induced model, as outlined in Section 7.1 with the sequentialization technique explained in Appendix C.2, we obtain the following gradient

---

[3]Code is available at https://github.com/uicdice/imm-reinforce

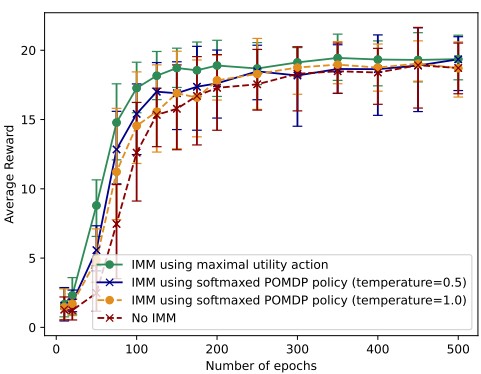

*Figure 9: Average reward of MDP trained without and with IMM incorporating POMDP solutions of various "qualities". Details in Appendix D.*

calculation for the IMM risk (compare to Eq. (20)):

$$\mathsf{IMM}_t(Q) = -\sum_y \overline{P}(y|\overline{x}_t) \log \hat{Q}(y|\overline{x}_t)$$

$$\nabla_W \mathsf{IMM}_t(Q) = -\sum_y \overline{P}(y|\overline{x}_t) \nabla \log \hat{Q}(y|\overline{x}_t)$$

$$= -\sum_y \overline{P}(y|\overline{x}_t) \nabla \log \left[ \sum_{t'=1}^n \left( \frac{w_{t,t'}}{\sum_{t'=1}^n w_{t,t'}} \right) Q(y|\overline{x}_t, \underline{x}_{t'}) \right]$$

$$= -\sum_y \overline{P}(y|\overline{x}_t) \frac{\nabla \left[ \sum_{t'=1}^n \left( \frac{w_{t,t'}}{\sum_{t'=1}^n w_{t,t'}} \right) Q(y|\overline{x}_t, \underline{x}_{t'}) \right]}{\sum_{t'=1}^n \left( \frac{w_{t,t'}}{\sum_{t'=1}^n w_{t,t'}} \right) Q(y|\overline{x}_t, \underline{x}_{t'})}$$

$$= -\sum_y \overline{P}(y|\overline{x}_t) \frac{\nabla \left[ \sum_{t'=1}^n w_{t,t'} Q(y|\overline{x}_t, \underline{x}_{t'}) \right]}{\sum_{t'=1}^n w_{t,t'} Q(y|\overline{x}_t, \underline{x}_{t'})}$$

$$= -\sum_y \sum_{t'=1}^n \overline{P}(y|\overline{x}_t) \frac{\nabla \left[ w_{t,t'} Q(y|\overline{x}_t, \underline{x}_{t'}) \right]}{\sum_{t'=1}^n w_{t,t'} Q(y|\overline{x}_t, \underline{x}_{t'})}$$

$$= -\sum_y \sum_{t'=1}^n \overline{P}(y|\overline{x}_t) \frac{\nabla \left[ w_{t,t'} Q(y|\overline{x}_t, \underline{x}_{t'}) \right]}{\sum_{t'=1}^n w_{t,t'} Q(y|\overline{x}_t, \underline{x}_{t'})} \frac{Q(y|\overline{x}_t, \underline{x}_{t'})}{Q(y|\overline{x}_t, \underline{x}_{t'})}$$

$$= -\sum_y \sum_{t'=1}^n \overline{P}(y|\overline{x}_t) \frac{w_{t,t'} Q(y|\overline{x}_t, \underline{x}_{t'})}{\sum_{t'=1}^n w_{t,t'} Q(y|\overline{x}_t, \underline{x}_{t'})} \frac{\nabla Q(y|\overline{x}_t, \underline{x}_{t'})}{Q(y|\overline{x}_t, \underline{x}_{t'})}$$

$$= -\sum_{t'=1}^n \sum_y \overline{P}(y|\overline{x}_t) \underbrace{\frac{w_{t,t'} Q(y|\overline{x}_t, \underline{x}_{t'})}{\sum_{t'=1}^n w_{t,t'} Q(y|\overline{x}_t, \underline{x}_{t'})}}_{\text{crosstalk}} \nabla \log Q(y|\overline{x}_t, \underline{x}_{t'})$$

$$\nabla_W \mathsf{IMM}(Q) = - \underbrace{\frac{1}{n} \sum_{t=1}^n \sum_{t'=1}^n}_{\text{double summation}} \sum_y \overline{P}(y|\overline{x}_t) \underbrace{\frac{w_{t,t'} Q(y|\overline{x}_t, \underline{x}_{t'})}{\sum_{t'=1}^n w_{t,t'} Q(y|\overline{x}_t, \underline{x}_{t'})}}_{\text{crosstalk}} \nabla \log Q(y|\overline{x}_t, \underline{x}_{t'})$$

Here $w_{t,t'}$ uses the index of $x_1$, rather than its value, as argument. By using a density estimator that looks at the entire dataset, we have a sampling scheme that effectively has set $k = n$, costing an $n$-fold computational time increase. While we can address this through shortcuts (e.g., sparsifying $w$), this extreme version of computing the induced model is revealing in terms of where the bottleneck is, i.e., computing $\hat{Q}$.

For the Serialized IMM variant, by comparing the above to Eq. (13), we see the large gain thanks to the double summation collapsing into a single summation. We still need to compute $\hat{Q}$ for the correction factor, but we can do this every $n$ iterations. By incurring the $\mathcal{O}(n)$ slowdown only $\frac{1}{n}$ of the time, we go back to only a constant factor overhead. The additional noise in the gradient and the fact that $\hat{Q}$ is stale between updates mean that Serialized IMM cannot be expected to perform at the same level as Sampled IMM. Furthermore, the Serialized IMM does not use the smooth nearest-neighbor averaging, *except* for the calculation of $\overline{Q}$. Despite this, we see in Figure 10 that Serialized IMM remains better than noising and, for larger dataset size, starts to approach Sampled IMM. Note that for ease of reproducibility, a fixed $\lambda = 0.7$ was used for these results and no particular optimization tweaks were made to compensate for gradient noise, suggesting that performance could be improved by making such changes.

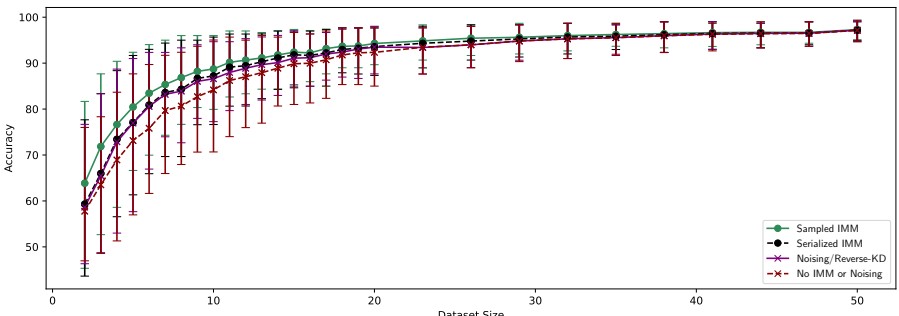

*Figure 10: Serialized IMM vs. Sampled IMM and Noising for logistic model. Serialized IMM provides $\mathcal{O}(k)$-times computational speedup over Sampled IMM (where $k$ is the number of samples and, in this case, $k = $ dataset size). The serialized version lacks the smoothing of the sampled version yet approaches it for larger dataset sizes and always improves on noising.*

## E  Potential Negative Societal Impact

Improving language models can have negative societal impact, if these language models are fine-tuned for tasks that are not aligned with positive human values. We hope that progress achieved by new methodologies such as IMM will not be put into misuse of this kind. On the contrary, we expect that any improved statistical efficiency achieved by methods like IMM to bring value when data is not as abundant, thus helping make machine learning and language modeling more impactful in underserved domains, where data is more scarce.

# NeurIPS Paper Checklist

1. **Claims**

   Question: Do the main claims made in the abstract and introduction accurately reflect the paper's contributions and scope?

   Answer: [Yes]

   IMM as a generalization of noising is analytically shown in the paper, along with noising's caveats that are solved by IMM. It's application is experimentally demonstrated in the paper, in Language Modeling and Reinforcement Learning experiments, as well as a proof of concept, using Logistic Regression.

2. **Limitations**

   Question: Does the paper discuss the limitations of the work performed by the authors?

   Answer: [Yes]

   IMM, in some applications like noising, requires random sampling, because exact evaluation is not feasible. This affects the accuracy of the induced model, but regardless, IMM is able to outperform the baseline (i.e. noising).

3. **Theory Assumptions and Proofs**

   Question: For each theoretical result, does the paper provide the full set of assumptions and a complete (and correct) proof?

   Answer: [Yes]

   The only proof is the one that discusses the caveat of noising through a counter example. All assumptions and a complete proof is present.

4. **Experimental Result Reproducibility**

   Question: Does the paper fully disclose all the information needed to reproduce the main experimental results of the paper to the extent that it affects the main claims and/or conclusions of the paper (regardless of whether the code and data are provided or not)?

   Answer: [Yes]

   Code for all experiments has been anonymously released with clear instructions for reproducibility. A summary of relevant experimental details is also contained within the paper.

5. **Open access to data and code**

   Question: Does the paper provide open access to the data and code, with sufficient instructions to faithfully reproduce the main experimental results, as described in supplemental material?

   Answer: [Yes]

   Code for all experiments has been anonymously released with clear instructions for reproducibility. A summary of relevant experimental details is also contained within the paper.

6. **Experimental Setting/Details**

   Question: Does the paper specify all the training and test details (e.g., data splits, hyperparameters, how they were chosen, type of optimizer, etc.) necessary to understand the results?

   Answer: [Yes]

   A summary of relevant experimental details is also contained within the paper.

7. **Experiment Statistical Significance**

   Question: Does the paper report error bars suitably and correctly defined or other appropriate information about the statistical significance of the experiments?

   Answer: [Yes]

   Error bars have been provided along with the number of Monte Carlo runs performed to obtain each bar.

8. **Experiments Compute Resources**

   Question: For each experiment, does the paper provide sufficient information on the computer resources (type of compute workers, memory, time of execution) needed to reproduce the experiments?

   Answer: [Yes]

   We mention the GPU used (Nvidia V100 with 32 GB memory)

9. **Code Of Ethics**

   Question: Does the research conducted in the paper conform, in every respect, with the NeurIPS Code of Ethics `https://neurips.cc/public/EthicsGuidelines`?

   Answer: [Yes]

   Improving language models can have negative societal impact, if these language models are fine-tuned for tasks that are not aligned with positive human values. We hope that progress achieved by new methodologies such as IMM will not be put into misuse of this kind.

10. **Broader Impacts**

    Question: Does the paper discuss both potential positive societal impacts and negative societal impacts of the work performed?

    Answer: [Yes]

    Appendix E addresses this.

11. **Safeguards**

    Question: Does the paper describe safeguards that have been put in place for responsible release of data or models that have a high risk for misuse (e.g., pretrained language models, image generators, or scraped datasets)?

    Answer: [NA]

    None of our datasets are scrapped. Pre-trained checkpoints are from reliable sources (Google trained checkpoint).

12. **Licenses for existing assets**

    Question: Are the creators or original owners of assets (e.g., code, data, models), used in the paper, properly credited and are the license and terms of use explicitly mentioned and properly respected?

    Answer: [Yes]

    All our code (both original and derivative) is Apache 2.0 licensed and a copy of this license is included in all code repositories. Datasets are public domain.

13. **New Assets**

    Question: Are new assets introduced in the paper well documented and is the documentation provided alongside the assets?

    Answer: [Yes]

    The only new asset is code. Code for all experiments has been anonymously released with clear instructions for reproducibility. A summary of relevant experimental details is also contained within the paper.

14. **Crowdsourcing and Research with Human Subjects**

    Question: For crowdsourcing experiments and research with human subjects, does the paper include the full text of instructions given to participants and screenshots, if applicable, as well as details about compensation (if any)?

    Answer: [NA]

    No human subjects were involved.

15. **Institutional Review Board (IRB) Approvals or Equivalent for Research with Human Subjects**

Question: Does the paper describe potential risks incurred by study participants, whether such risks were disclosed to the subjects, and whether Institutional Review Board (IRB) approvals (or an equivalent approval/review based on the requirements of your country or institution) were obtained?

Answer: [NA] .

No human subjects were involved.

