# OpenReview forum: "Induced Model Matching: Restricted Models Help Train Full-Featured Models"
_NeurIPS.cc/2024/Conference — NeurIPS 2024 spotlight_

### Official Review · Reviewer_vLY4 · 2024-06-13

**Soundness:** 3
**Presentation:** 4
**Contribution:** 2
**Rating:** 6
**Confidence:** 4

**Summary:**

This paper proposes a method to train models which have access to all of the features by making their marginal distribution match a known weaker model which only uses a subset of those features to predict. The authors relate this to knowledge distillation and noising, and come up with an approximate objective which computes this marginalization over the bigger model. Even though it might be intractable to do analytically, they provide a sampling-based method for the computation. They evaluate the model on logistic regression and language modelling

**Strengths:**

- The relation of the proposed method to the previous work in noising is insightful and not immediately obvious, giving an interesting perspective.
- The connection to knowledge distillation is valuable to highlight.
- The code for the method is available.
- The text is generally very well-written and explains the proposed method very well.

**Weaknesses:**

- The empirical results are somewhat dated. The models considered are quite old (RNNs, BERT) and it would be nice to see a more modern set of models such as decoder-only transformers or state-space models. However, using the old models does allow comparison to the noising approach. While the direct comparison to noising is appreciated, in practice most people would likely be comparing to the vanilla training approach as a baseline.

- The main weakness of the paper is the computational overhead incurred by this method, which is not adequately addressed in the paper. With the introduction of language modeling, the question of how this method would scale to larger datasets and models is important, and the scaling properties do not appear favorable.
- For the RNN experiment, each forward and backward pass seems to require k additional passes due to the gradient accumulation for the samples drawn for the model matching objective. This incurs a significant overhead factor of k.
- The overhead factor may actually be worse when considering the need to replace the context for each of the k summands in the IMM objective. This requires swapping out the current context and computing a new set of hidden vectors, potentially up to the length of the input. A back-of-the-envelope calculation suggests the FLOPs could increase from O(L) to O(L + kL^2). However, there may be some subtlety I'm missing here that reduces the overhead.
- For a large dataset, the lookup of the context could be quite difficult to implement. In the worst case, for n sequences of length L, storing the auxiliary data structure could require space on the order of nL^2, which is a factor of L larger than the original dataset. Supporting random access to this data structure may not be feasible for large corpora like Common Crawl without significant infrastructure overhead.

**Questions:**

My questions all relate to my understanding of the limitations of the IMM approach.  I would be happy to reconsider my assessment if the above questions are addressed satisfactorily.

1. Could the authors clarify the approximate computational and memory complexity of this method in terms of the number of sequences n, sequence length L, and any other relevant parameters? It would be helpful to see the analysis for both RNNs and Transformers.

2. Could the authors comment on the asymptotic space complexity of the auxiliary data structure? For large corpora, it seems like this could require a huge amount of disk space and be challenging to query efficiently. Is there a data structure that could mitigate these concerns?

3. Could the authors provide a FLOPs-corrected comparison between the IMM method and the vanilla training approach? Given the higher computational cost per forward and backward pass for IMM, it would be informative to see how the two methods compare when given a similar computational budget.

4. Can the authors comment on practical cases where the IMM approach would be preferred over spending the additional $k$ FLOPS on e.g. processing $k$ more examples?

**Limitations:**

Yes

---

> ### Author Rebuttal · Authors · 2024-08-07
>
> Thank you for your deep reading of our paper and for being open-minded about revisiting your assessment! Regarding the key points you raise:
>
> + **Empirical results dated** — You are correct that the motivation for comparison with LSTMs was because of  the noising baselines existing there. We used the encoder/decoder BERT to illustrate validity for Transformers, however we didn't have the infrastructure for large-scale models like LLMS. We hope that the body of all the experiments we've included, non-language models too, show the wide merit of the approach.
>
> + **Computational overhead** — IMM, as implemented in the paper, does have a computational cost relative to the baseline of training only with cross-entropy. We detail this overhead in answering your questions below. You are right that this is a limitation, however our focus was on the statistical aspect. Given limited data (so additional samples are not an option) along with a good restricted model (potentially derived from the same data, or from a corpus of restricted data), IMM achieves improved accuracy/perplexity/reward. That said, in the general rebuttal we outline a potential solution to tackle the computational overhead.
>
>
>
> Regarding your questions:
>
> First, to clarify, your use of $L$ should be replaced with the unroll length of the LSTM, which is the maximum depth through which gradients are backpropagated (set to 35). Also, to be consistent with the paper, we use $n$ for the length of the data (in tokens/words).
>
> 1. **Time complexity** — The best way to quantify the computational overhead is relative to the baseline of using traditional cross-entropy.
>   +  Let $L$ again represent the unroll length of the LSTM, then since histories have to be swapped at every unroll location and the LSTM re-evaluated, the overhead factor is $\mathcal{O}(kL)$. Therefore your expression is partly correct, apart from $L$ representing unroll length and $n$ data size, i.e. we go from $\mathcal{O}(n)$ to $\mathcal{O}(nkL)$ per epoch. We partially mitigate this in our current implementation by applying IMM periodically (not at every iteration), see Appendix D2.2. For example, if we do it only every $\Omega(1/L)$ iterations, the overhead factor becomes $\mathcal{O}(k)$ and we go from $\mathcal{O}(n)$ to $\mathcal{O}(nk)$ .
>   + For BERT/Transformers (and in fact in our non-language modeling experiments), the baseline does not have a recurrence that require new passes, and therefore the overhead is $\mathcal{O}(k)$, so we go from $\mathcal{O}(n)$ to $\mathcal{O}(nk)$.
>   + A $k$-fold increase is not ideal, but it is acceptable as our choices of $k$ range from $5$ to $10$.
>
> 2. **Space complexity** — There are two overheads:
>   + *Model memory*: As explained in Appendix C.2, we sequentialize the gradient computation across the $k$ sample. This means that the space overhead during training is a factor of $2$ compared to the baselines (i.e., a second set of gradients), which are themselves of the order of the number of parameters of each model.
>   +  *Lookup overhead*: You are correct that a naive implementation of the data structure could take space $\mathcal{O}(nL)$. However, the better implementation is with a dictionary/hash table of lists containing indices/pointers to a reverse linked list representing the data set. By referencing a position, we can recover a length-$L$ history at any position, at runtime. This requires space $\mathcal{O}(n)$ only,  because each key in the dictionary is a short history, and the number of long histories is the same as the number of times that short history appears. Adding up the number of occurrences of all short histories, gives us $n$.
>
> 3. **FLOPS-Correction**  and 4. **Comparison** — For our reply, we would like to address an important misunderstanding here.
>   + IMM is not expected to be used in situations where we have access to an interminable data stream, where we can simply continue to train the cross-entropy baseline until the clock runs out. Even in the best of cases (see the general rebuttal) IMM could take twice as much time as the baseline, and it's unclear when the performance gain could exceed having twice as much data (often the gains seem to be equivalent to 20-30% more data.)
>   + Rather, IMM is expected to be used when the data is what it is, and one is required to additionally incorporate the side-information presented in the form of the feature-restricted model. The goal is to obtain a statistical advantage, which IMM does, at the expense of computational overhead. In the premise of IMM, it simply is not an option to collect that extra full-featured data. Therefore FLOPS-corrections and comparisons are not germane to the context, and if done they will unfairly show that IMM is not competitive. We hope that you do accept this explanation as to why this request cannot be addressed equitably.
>
> We believe that we have addressed all your major points. If there is anything else we could elucidate, please don't hesitate to ask us during the discussion period. We appreciate your deep insight about the computational aspect of the problem. We hope that we have quantified our current overhead and given insight about how this overhead can be reduced in the future. However, considering that our focus was primarily statistical, we hope that you will judge the merits of the paper on that basis. We believe we have something very valuable for the community. We would immensely appreciate it if you could recommend acceptance of the paper! Thank you once again for all your time and effort.

---

> > ### Comment · Reviewer_vLY4 · 2024-08-12
> > **Response**
> >
> > Thanks for the detailed response.
> > After reading the rebuttal and other rebuttals and reviews, I am still a bit skeptical of the computational feasibility of the approach. Slowing down training by a factor of 5-10 is quite a sizeable decrease, and this should be addressed in the main text and not glossed over. That said, I appreciate the authors' willingness to engage with the topic in the rebuttal and especially to identify alternative, equivalent, objectives that are more tractable (such as in the main response).
> >
> > As the authors point out, I understand the focus is on the relatively small-data regime, where IMM does seem to show some promise. I will raise my recommendation, although I urge the reviewers to try their best to make the IMM approach more computationally tractable for their next revision if they want it to be adopted more widely.

---

> ### Author Response · Authors · 2024-08-14
> **Thank you so much! + Update on a computationally efficient version.**
>
> Dear Reviewer,
>
> We are humbled by and very appreciative of your decision to move our work firmly into the accept zone. Thank you so much! Computation is not an afterthought for us. We will itemize both time and memory complexity in detail in the paper, just as we did in our rebuttal.
>
> More importantly, to show our dedication to making IMM computationally efficient, we were working very hard on implementing the computationally efficient version that we proposed in the general rebuttal. We had to surmount several technical hurdles, but we were finally successful in applying it to the logistic regression example. In a comment to the general rebuttal, we report on these results. The new method is only 50-70% slower than the baseline, but it continues to have an edge on noising and tracks the version of the paper very closely, especially as the dataset size increases. We also highlight how the same steps could be taken in the language modeling examples.
>
> We don't claim to have fully solved the problem, but we now have both a formulation for computationally efficient IMM and a proof of concept. We plan to describe these in the current paper and to add it to our published code. Thank you very much for pushing us to do this. The case for IMM is so much stronger as a result.
>
> You've been very generous with your revised opinion and we hope that you will continue to support the paper. (We have revised this paper a few times, and it would be great if our final revision is for the camera-ready version at this year's NeurIPS.)
>
> Thank you again for everything.
>
> Sincerely,
> Authors

---

### Official Review · Reviewer_ADbo · 2024-07-10

**Soundness:** 3
**Presentation:** 2
**Contribution:** 2
**Rating:** 6
**Confidence:** 3

**Summary:**

The paper considers the learning problem when in addition to the training set an additional _restricted_ model is available. The restricted model is trained on a different dataset, potentially containing a only subset of features. It is proposed to augment the training loss with the special induced model matching (IMM) loss that encourages similarity between full and restricted predictive distributions where the extra features are marginalized on the empirical data distribution. IMM is compared to similar techniques such as noising and weak-teacher knowledge distillations where the benefits of IMM are demonstrated theoretically. Experiments are conducted on a toy logistic regression problem, language modelling tasks and reinforcement learning in a toy environment.

**Strengths:**

The proposed idea is interesting and it seems to be an improvement over the existing techniques such as noising and reverse-KD. I see value in being able to systematically incorporate restricted knowledge from a different distribution. Empirically IMM seems to be helping on a number of LM tasks.

**Weaknesses:**

The paper adopts rather exotic notation that doesn't help reading the paper. I had to constantly look into the glossary in the appendix to go through equations. Descriptions of each experiment are also not self-contained, even when reading the appendix. The general idea of IMM is not the most intuitive to me and I think the paper would benefit from a simple graphical illustration of the principle. In my opinion, insufficient clarity in presenting the method and describing experiments is the main weakness of the paper.

If I understand the setup of LM experiments correctly (full and restricted models are trained on the same dataset) then the idea of the restricted model capturing a richer distribution is under-explored and the only way I can see IMM improving performance is as a regularizer (see my questions).

**Questions:**

1. I struggle to find a clear intuitive explanation for why induced model matching helps. Can authors think of a simple 2-dimensional logistic regression problem (for example) and include a graphical illustration of full and restricted models (and their induced variants)?

2. Do I understand it right that the LM experiments the restricted n-gram model has been "trained" exactly on the same dataset on which the full model is being trained?
2.1 If so, then, again, the only reason why IMM helps I can think of is some kind of "smoothing" or regularization of the full model, which probably learned an overly "kinky" distribution from limited data. In the eyes of the authors is that the right intuition?

3. Is there an experiment where the restricted model indeed comes from a different and much larger dataset (which authors mention as indeed an interesting scenario for applying IMM)? It would be interesting to look at the "trade value" of a"full" data point compared to a"restricted" data point.

4. Could authors hint at the scenarios in which IMM is not useful or even harmful? I would be especially interested if that could be the property of the model classes or data distributions from the full and the restricted model come.

**Limitations:**

Adequately discussed.

---

> ### Author Rebuttal · Authors · 2024-08-07
>
> Thank you for your very constructive review! We address some of the points raised:
>
> + **Notation** — Thank you for your suggestions on improving the notation. We have been working on a few potential alternatives to further clarify our presentation. Here's what we suggest to make things more readable.
>   - Instead of using $y_{-}$ or $y_{-t}$ for context, we'll use $x$ nd $x_t$, and reserve $y$ and $y_t$ only for predictions.
>   - In order to denote the short/extended components of a context, instead of using $\textsf{sh}(x)$ and $\textsf{ex}(x)$, we'll use $\overline{x}$ and $\underline{x}$ respectively. This will parallel using $\overline{P}$ when denoting the induced model, which only depends on the short history.
>   - While we tried to be very explicit in our notations, we believe these changes will declutter equations and make everything more legible. We hope that you approve. We are open to other suggestions.
> + **Graphical Representation** — Thank you for the suggestion. We propose to offer a high-level visual representation of the approach. [We have drafted the following figure](https://i.imgur.com/NwQgOPV.png), which also incorporates the notational changes suggested above. We hope to improve this and include it in the paper.
> +  **Experiment Descriptions** — In the main body of the problem, it is difficult to include all the details of the experiments. Appendix D has all the details. However, we will revisit how we have divided this information across the main body and the appendix. We will make every effort to make each experimental setup self-contained in the main body, with precise references to the appendix.
>
>
> Also your questions:
>
> 1. **Visualizing restricted model in logistic regression** — [The following figure](https://i.imgur.com/G4lLm2u.png) illustrates the 3-dimensional logistic regression problem in the paper. The features are samples uniformly in this box. The Bayes-optimal restricted model only uses the x-coordinate, and so assigns probabilities proportionally to the blue/red areas in the illustrated slice. IMM then encourages the full logistic model to be consistent with these weights, i.e., making sure the proportion of points labeled $\pm$ agrees with these weights at each x. Intuitively, this biases the separating plane to have the right inclination/alignment with the x-axis, which subsequently speeds up the learning process.
> 2. **When the data provides the restricted model** — Your intuition is correct for our own language modeling experiments. We do use the data itself to create our bigram for PTB, and for BERT we use Wikipedia, which it is pretrained on. The extensive literature that tries to incorporate N-grams into more sophisticated language models argues indeed that N-grams tend to capture structural detail that may be missed by the larger model, albeit in their restricted setting. However, some newer papers train N-grams on much larger datasets (since it's faster to do so), which adds more benefit (see Liu et al. 2024).
> 3. **When the restricted model comes from elsewhere** — In our other experiments, logistics regression and RL, the restricted model is more powerful/rich. As mentioned above, for logistics regression it is the Bayes-optimal restricted model. For RL, it is the exact solution for the POMDP, which can be thought of as full exploration of the reward landscape, but with only one-coordinate observation. We do not have a data-point by data-point value comparison, however, we do have in Appendix D4 (Figures 6 and 7) the effect of artificially weakening those perfect models. The bottom line is that as long as we have that extra data to build a decent approximation for the true restricted model, then IMM improves performance, in a way commensurate with the quality of the model (thus the amount of restricted data.) We have not done a theoretical analysis of this tradeoff, but it's an excellent suggestion for future investigation.
> 4. **When is IMM harmful** — In our experience, we can identify two scenarios in which IMM can be harmful:
>   + If, simultaneously, (a) the restricted model is of bad quality, and (b) the value of $\lambda$ is not properly tuned. This is hinted at in the importance of tuning lambda properly in the low-quality model experiments of Appendix D4. Therefore, you are right that if the model class of the restricted model is not powerful enough, it may not capture the true restricted model properly, and could lead to harm. (However, this goes against the main premise of the paper, i.e., the availability of good target models)
>   +  Also, when the amount of data is so limited or the distribution of the extended context given the short context is hard to learn (e.g, lack of structure such as smoothness or latent low-dimensionality), then the learned induced model $\hat Q$ will not be accurate, and the performance can suffer. We see a hint of this in Figure 1, where with very few data points IMM is bested by noising (though it is not worse than no-IMM).
>
> We did our best to address your concerns. If there is anything else we could elucidate, please don't hesitate to ask us during the discussion period. We appreciate your leaning toward acceptance. We hope that you will engage us further, and that based on this we will earn a higher score from you! Thank you so much for your time and effort.

---

> > ### Comment · Reviewer_ADbo · 2024-08-12
> >
> > I thank authors for their clarifications and I'm raising my score for 1 point. I hope that the improved notation and the toy task figure (which I would still improve) will make it into the future version of the paper.

---

> > > ### Author Response · Authors · 2024-08-14
> > > **Thank you very much!**
> > >
> > > Dear Reviewer,
> > >
> > > We thank you very much for approving of the new notation. We will definitely use it in the paper, since it's so much more streamlined. We also have a few ideas on how to improve the visualization of the toy example. In particular, we can display the Bayes restricted model and a model-in-training along with its induced model, and show how IMM encourages one to tend toward the other (the alignment that we mentioned).
> > >
> > > We appreciate a lot the additional point to your score! We're sorry for not replying sooner. We were spending a lot of time researching a computational efficient variant of IMM. We were finally able to implement it for the toy example, and our results are summarized in a comment to the general rebuttal. Please feel free to consult it as you make your final recommendations.
> > >
> > > Sincerely,
> > > Authors

---

### Official Review · Reviewer_2S2s · 2024-07-12

**Soundness:** 3
**Presentation:** 3
**Contribution:** 3
**Rating:** 6
**Confidence:** 4

**Summary:**

The authors introduce the problem of “Induced Model Matching” where there exists a small and restricted model that only takes into account some of the features and is able to predict relatively well the label given these features. The key question of this paper is how one can leverage such a small model when training a full-feature, larger model. The authors answer this question by providing an algorithm which aims to match the restricted features version of the large model to the restricted features model. The authors suggest a regularization term to the loss, called IMM, which accomplishes this. The authors present a toy experiment of regression as well as a larger-scale experiment with language, showing that their method is able to perform better when using the IMM regularization.

**Strengths:**

* This is a very interesting topic, and seems like it could be useful in real-world scenarios. The idea appears novel and also is an interesting angle when compared to knowledge distillation.
* It is very well-analyzed, and thoroughly explains how the method compares to existing methods.
* The authors provide a clear presentation of their ideas as well as clear definitions. It is also nice to also have included a glossary in the appendix. Overall the paper is very clear to follow and precise about definitions.
* The experiments show that, indeed, the proposed regularization term benefits the large model.

**Weaknesses:**

* As the authors mention, it is difficult to compute the regularization term. The computational cost is a drawback, although having increased performance is still a nice result. Given the computational cost, this brings into question when this method becomes useful in practice. I will leave my thoughts on this in the questions section.
* It would be interesting to include further analysis regarding how the restricted model’s performance affects this method.

**Questions:**

* It might be realistic to assume that $\hat{P}$ has some loss $\varepsilon$. Do you have any insights  as to how one might expect IMM to degrade as a function of $\varepsilon$?
* Did you consider the setting in which there are many restricted feature models? How do you think your setting can be generalized even further? Do you think there are any settings in which using IMM regularization might hurt the model? For example, if the task is recalling something from a long time ago in the context, it might seem reasonable to assume this method would not work. Generally speaking, what kinds of problems do you think your method is best suited for? Information theoretically, there might be examples in which having some base solver might make the overall problem easier, but as mentioned, there might also be scenarios in which this method actually harms the training of the overall model.
* Just for my understanding, are you assuming a fixed-context length model?

**Limitations:**

Yes.

---

> ### Author Rebuttal · Authors · 2024-08-07
>
> Thank you very much for deeply understanding and appreciating our paper. We address some of the points raised:
>
> + **Computational cost** — Our general attitude is: given a fixed amount of data and a feature-restricted model, how can we do most with it? We are thus mostly concerned with statistical performance, however we acknowledge that computational performance is critical for applicability. In the general rebuttal, we propose approaches to more radically overcome computational hurdles in the future.
> + **When $\hat P$ is of lower quality / how this quality affects things** —  We address this experimentally. Due to lack of space, we moved this to Appendix D4 (Figures 6 and 7). There, we artificially weaken the models in the logistic regression and RL experiments. (Additionally, the Kneser-Ney bigram in language modeling is very good, but it certainly is not the ideal bigram of English. We could also do an ablation by worsening it.) The takeaway is that, by tuning $\lambda$, IMM always helps and never hurts, even with good but suboptimal restricted models, and its gains  are commensurate with the quality of the model.
> > We do not yet have a theoretical analysis of restricted-model quality vs. benefit. However, the insight of why this works is that the dual interpretation of $\lambda$ is as constraining the learned model to have its induced version $\overline{Q}$ in a $\delta$-ball around $\hat P$, with larger $\lambda$ meaning smaller $\delta$. If $\hat P$ is $\epsilon$-away from the true $\overline{P}$, we need to make $\delta\geq \epsilon$, to make sure that we're not harming the learned model by keeping it artificially away from the truth. When the model is high quality ($\epsilon$ small), we can thus make $\delta$ small, and thus $\lambda$ can be large. However, when the quality is low ($\epsilon$ large) then we need to make $\delta$ large too, which means that $\lambda$ has to be small, and therefore the relative benefit from IMM diminishes.
> + **Did you consider many restricted-feature models** — We did think about this, but we haven't worked on it yet. Similarly to multi-task learning, we imagine that adding multiple IMM losses, each for a different restriction, could be able to handle this.
> + **Other generalizations** — One setting we contemplated to try IMM in is in building physics-informed models. A closed-form physics model (e.g., climate prediction) can be thought of as a restricted model, because it often relies on very specific features. A full-featured model could use many more features than the physics model. IMM can be used as a method to incorporate the physics model, by making sure that the full-featured model's induced version matches the physics model. Another generalization that we're studying is to take IMM beyond feature-restriction, to also cover task-restriction: incorporating models that can perform a sub-task very well.
> + **Settings where IMM would not help** — As we have noted in Appendix D4, if the restricted model is bad and $\lambda$ is not tuned properly, IMM could hurt. However, let us assume that the restricted model is good, perhaps even the ideal restricted model. In this case, we speculate that IMM can only help. It may be counterintuitive, but even if the restricted features are uninformative for the task (as in your example), IMM can help. Indeed, if the task is impossible to perform with the restricted features, then the best restricted model will be no better than chance. IMM will try to make the learned induced model mimic this, and perform no better than chance too. We believe this will have the effect of informing the full model's training that the restricted features are pointless, which is useful information! Without this guidance, the full model could waste resources/data to discover this fact. In contrast, with this guidance, it can search the subset of hypothesis space that does not use the restricted features. As this is a smaller space, IMM can be thought of as regularizing the learning even in this case. Therefore, we believe our method is suited for any problem in which we can have a reasonably  accurate approximation of the restricted model.
> + **Are we assuming a fixed context length model** — The full context can have variable length, however, we are assuming that the short context has a fixed length. That said, this is mostly about how we're formalizing this in the paper. The methodology itself can work with either, as long as there is a consistent way to split each full context into a short and extended context.
>
> We hope these address your points. If there is anything else we could elucidate, please don't hesitate to ask us during the discussion period. We know you've already been very generous with your score, but if you could consider a higher score, AC's decision would be easier, and we would be extremely grateful! Thank you again for your time and effort.

---

> ### Author Response · Authors · 2024-08-14
> **Did we address your points?**
>
> Dear Reviewer,
>
> Thank you again for taking the time to review our paper. We understand that life is busy and that reviewing can be hectic. We believe that we addressed all the points that you raised.
>
> Additionally, we would like to bring your attention to the fact that (in the general rebuttal) we proposed a formulation for making IMM computationally efficient, and that we just finished implementing it successfully for the logistic regression experiment. We hope that you will also take that into consideration when making your final recommendation.
>
> If we did address all your points, we very much hope that you will raise your score to reflect it. It would helps us get closer to sharing this work with the community and we would appreciate that greatly!
>
> Thank you again.
>
> Sincerely,
> Authors

---

### Official Review · Reviewer_vmaB · 2024-07-16

**Soundness:** 4
**Presentation:** 4
**Contribution:** 3
**Rating:** 7
**Confidence:** 3

**Summary:**

- Algorithm: This paper proposes a framework for how a good but restricted feature model, e.g. $\bar{P}(y \mid x_1)$, can be used as guidance when training a full-feature model, e.g. $Q(y \mid x_1, x_2, x_3...)$.
\
Instead of the knowledge distillation objective from weak teachers, which directly adds a regularizer comparing the probabilities of the restricted feature model \bar{P} and full-feature Q with each other directly, IMM proposes comparing P with Q marginalized over the other features
\
$$\mathsf{Reverse-KL}: \lambda \sum_{y-} \pi(x_1, x_2, x_3) \sum \bar{P}(y \mid x_1) \log \frac{1}{Q(y \mid x_1, x_2, x_3)}$$
$$\mathsf{IMM}: \lambda \sum_{y-} \pi(x_1, x_2, x_3) \sum \bar{P}(y \mid x_1) \log \frac{1}{\hat{Q}(y \mid x_1)} \text{ where } \\
\hat{Q}(y \mid x_1) = \sum_{x_2, x_3} \pi(x_2, x_3 \mid x_1) Q(y \mid x_1, x_2, x_3)
$$
- The marginalization can be quite expensive in practice, but they demonstrate (in small RL and Language tasks) that they are able to approximate the marginalization efficiently by sampling. The single-sample IMM objective is connected to noising/reverse-KD.
- They mathematically show that in the infinite data regime with the perfect true distribution $\bar{P}(y | x_1)$ and the true model P(y \mid x_1, x_2, x_3) is in the hypothesis class of Q, IMM objective always recovers P. On the other hand, reverse KL and noising cannot.
- Empirically, IMM always does better than with no-IMM, but the performance difference is especially large in data-limited regimes.

**Strengths:**

While the objective may still be too computationally expensive for very large-scale tasks (i.e., sufficiently sampling a suffix tree, high inference cost), the paper is thought provoking, and well-written/motivated. I would recommend it for acceptance.

**Weaknesses:**

- One weakness of the paper is obviously the small scale of most of the experiments (learning policies over 11x11 grid, subset of GLUE, toy linear model). But the experiments sufficiently demonstrate the proof of concept.
- It would also be nice to include some settings where the feature-restricted model is not close to optimal as an ablation study.
- In Figure 1, why is noising able to achieve almost the same performance as IMM starting from around 5% dataset size, if noising is similar to  single-sample IMM? Generally, what is the importance of using more samples when marginalizing Q for improved performance? Does performance strictly increase with increasing samples?

**Questions:**

-  Is there any practical importance to the infinite-data regime analysis? Are there circumstances where training with the IMM objective harms performance compared to just cross-entropy with increasing data?

---

> ### Author Rebuttal · Authors · 2024-08-07
>
> Thank you very much for deeply understanding and appreciating our paper. We address some of the points raised:
>
> + **Scale of the experiments** — The computational overhead of the model and the fact that we didn't have the infrastructure for large models, meant that we dedicated ourselves to simple reproducible benchmarks to demonstrate the merit of IMM. (Many of the experiments stem from our original goal of understanding the underpinnings of the practice of noising.) In the general rebuttal, we propose approaches to more radically overcome computational hurdles in the future.
> + **Suboptimal feature-restricted models** — We address this! We do it experimentally in Appendix D4 (Figures 6 and 7), where we artificially weaken the models in the logistic regression and RL experiments. (Additionally, the Kneser-Ney bigram in language modeling is very good, but it certainly is not the ideal bigram of English.) The takeaway is that, by tuning $\lambda$, IMM always helps and never hurts, even with good but suboptimal restricted models, and its gains  are commensurate with the quality of the model.
> + **Noising vs IMM at small dataset sizes** in Figure 1 — High level explanation: In this regime, IMM needs to marginalize/induce the model accurately and it can't do that at _extremely_ small data sizes. Lower-level explanation: although noising is a biased version of IMM, it also has less variance, because it doesn't need the induced model. With very little data,, that's useful. This explanation also suggests a possible best-of-both-worlds solution, by performing a bias-variance tradeoff. This is something we've thought about but not yet investigated since the issue only surfaces at such extremes. The performance of IMM increases with more samples for marginalization, because the variance is reduced. However, the returns diminish, and we find that k=5 to 10 samples are sufficient to get good performance.
> + **Infinite data analysis** — This analysis is just an analytical tool to contrast IMM and noising, and to reveal the latter's shortcomings.  If one truly had infinite data, neither of those techniques would be necessary. The only circumstance where IMM will hurt is if the target model is very wrong and the $\lambda$ is not tuned to account for it. (If the target model is correct, $\lambda$ can be arbitrary.) In contrast, the only circumstance where noising/reverse-KD will _not_ hurt is if the target model is correct and the $\lambda$ shrinks with more data. This is what we do in Figure 1 (i.e., for noising, we use the high quality feature-restricted model and we find the best $\lambda$ at each data size). Despite this, the benefits of noising disappear quickly with increasing data size.
>
> We hope these address your points. If there is anything else we could elucidate, please don't hesitate to ask us during the discussion period. We know you've already been very generous with your score, but if you could consider a higher score, AC's decision would be easier, and we would be extremely grateful! Thank you again for your time and effort.

---

> ### Comment · Reviewer_vmaB · 2024-08-10
> **Thanks, one more question**
>
> Thank you for the clarification! I think there was a misunderstanding regarding my question about Noising versus IMM.
>
> The authors provide intuition for why Noising does better than IMM at small dataset sizes, but rather my question was why the gap between Noising and IMM closes so quickly. By 5+% dataset size, the performance gap seems to have closed to around 3%, but my understanding is that IMM is much more computationally expensive. I wonder whether there are circumstances where IMM has any gains over cheaper methods trained on a little bit more data or whether IMM generally has any utility in any settings with reasonable amounts of data. I'm also not certain how the author's explanation aligns with Figure 1 observations. Noising seems to strictly be doing worse than IMM for any dataset size smaller than 5% and there seems to be no dataset size where IMM does worse than Noising.
>
> I will most likely keep my score as 7, if the authors would like to devote more time to answering other reviewers' questions.

---

> > ### Author Response · Authors · 2024-08-12
> > **Answer (1/2)**
> >
> > We misunderstood your question to refer to the only statistical advantage that noising has, which is smaller variance at small data sizes. We sincerely apologize for this. We now address all your related questions, point by point.
> >
> > >  In Figure 1, why is noising able to achieve almost the same performance as IMM starting from around 5% dataset size, if noising is similar to single-sample IMM?  / [...] my question was why the gap between Noising and IMM closes so quickly. By 5+% dataset size, the performance gap seems to have closed to around 3% [...]
> >
> > The gap narrows in Figure 1 because we are being very favorable to noising. Specifically, we are decaying $\lambda$ (the amount of noising) optimally with increasing data. This is _necessary_ for noising. The reason for this is the same as what occurs in Proposition 5.1: even if the target model is perfect, because noising incorrectly tracks the target model, without decaying its influence it will not only not narrow the gap, but would in fact derail the learned model. Decaying $\lambda$ is also acknowledged as critical in the reverse knowledge-distillation literature (see for example Sec. 3 of Qin et al. 2021). However, tuning $\lambda$ is _optional_ for IMM, thanks to $\hat Q$ (with more extended history samples) accurately tracking the target (see the flat curves on the right column of Figure 4 in Appendix D1.1).
> >
> > To fully convince you, we re-ran the experiment with fixed $\lambda=1.5$ (optimal at data size 5). The results are below. IMM maintains performance comparable to Figure 1, whereas noising experiences a widening gap, and soon underperforms even the baseline.
> >
> > | Dataset Size |   Baseline           | Noising             | IMM                 | IMM-Noising Gap |
> > |--------------|----------------------|---------------------|---------------------|-----------------|
> > | 5          | 73.14 +16.17/-14.53  | 76.85 +18.58/-13.15 | 79.96 +14.99/-12.38 | 3.11            |
> > | 10         | 84.17 +13.51/-10.86  | 84.37 +7.40/-7.63   | 88.65 +9.65/-7.68   | 4.28            |
> > | 15         | 89.86 +8.86/-6.80    | 86.17 +6.17/-5.53   | 92.52 +6.19/-4.81   | 6.35            |
> > | 20         | 92.35 +7.35/-5.32    | 86.99 +5.69/-5.37   | 94.30 +4.63/-3.70   | 7.30            |
> > | 30         | 94.94 +3.97/-3.43    | 88.68 +4.35/-4.32   | 95.69 +3.39/-2.64   | 7.01            |
> > | 40         | 96.33 +3.00/-2.33    | 89.70 +3.73/-3.97   | 96.59 +2.59/-2.08   | 6.89            |
> > | 50         | 97.14 +2.47/-2.20    | 90.78 +3.78/-3.55   | 97.34 +2.34/-1.99   | 6.56            |
> >
> > > Generally, what is the importance of using more samples when marginalizing Q for improved performance? Does performance strictly increase with increasing samples?
> >
> > The above answer gives one clear benefit of a more accurate $\hat Q$: less sensitivity to tuning $\lambda$. However, since tuning $\lambda$ is possible, the primary advantage is significant and consistent gains, even against optimally-tuned noising. The gains in Figure 1 may seem small in terms of absolute percentages, but they are significant, as the IMM-noising gap is larger than the noising-baseline gap, most of the time and often considerably so. We can quantify these gains in terms of data size increase: e.g., the accuracy of IMM at 15, would need augmenting the data set size to 18 with tuned noising (a 20% increase) or to 20 without noising (a 33% increase). These gains are also more consistent, as the variance of IMM's accuracy is smaller than the variance of noising's accuracy, most of the time (except, as noted, for small data sizes).
> >
> > IMM strictly improves when we use more extended history samples, but with diminishing returns, allowing us to cap to $k=10$ samples at most.

---

> > > ### Author Response · Authors · 2024-08-12
> > > **Answer (2/2)**
> > >
> > > > [...] but my understanding is that IMM is much more computationally expensive.
> > >
> > > We address the computational aspect of IMM in the general rebuttal. In essence, the alternative that we propose has the potential to make IMM computationally equivalent to noising.
> > >
> > >
> > > > I wonder whether there are circumstances where IMM has any gains over cheaper methods trained on a little bit more data [...]
> > >
> > > Our general premise is that we have what data we have, and no more. Noising and IMM both have this data and the same feature-restricted) target model. (The sampling to estimate $\hat{Q}$ is done with the same data set.) With these exact same information resources, IMM always statistically outperforms noising. If the goal is to make the most out of what information we have, or when acquiring that little bit more data is expensive, IMM is the way to go.
> > >
> > > > [...] or whether IMM generally has any utility in any settings with reasonable amounts of data.
> > >
> > > This question can be asked of any variant of data augmentation, as given enough data there is no need to augment it. The question is: in what regime is any such technique useful? In Section 6.1, we mention that IMM appears most successful when data size is comparable to the number of parameters. This agrees with the general rule of thumb of regularization benefiting high-dimensional regimes. The intuition is that the cross-entropy loss does not sufficiently localize the model, so we get better localization when IMM constrains the induced model, which is a low-dimensional projection. Since most modern models (even LLMs) operate in such regimes, it is not a stretch to expect IMM to be widely beneficial. (Otherwise, it would be also hard to imagine the many techniques suggesting using N-grams to augment LLMs having any utility either.)
> > >
> > > ---
> > >
> > > We are convinced that the new perspective that IMM provides, the deeper understanding of noising and reverse knowledge-distillation that it offers, and the consistent statistical edge that it has with limited data, make it worthwhile to share with the community at NeurIPS. Once the idea is out there, we are certain that our effort and that of others can inevitably make IMM more computationally efficient and scalable.
> > >
> > > You have been very generous with your assessment of our work. If you believe 7 is the right score, we humbly thank you and only hope that your position can help convince your colleagues to also support us at the same level.

---

> > > > ### Comment · Reviewer_vmaB · 2024-08-14
> > > > **Thanks!**
> > > >
> > > > Thank you for the thorough clarification! My questions have been answered. I leave my final score as a 7, after consideration. The paper is well written and proposes interesting ideas and algorithms. In the future, if authors can demonstrate the usefulness of this method on more timely larger-scale applications, I think that could make the paper even stronger.

---

> > > > > ### Author Response · Authors · 2024-08-14
> > > > > **Thank you!**
> > > > >
> > > > > Dear Reviewer,
> > > > >
> > > > > Thank you very much for being very vocal about the qualities of our paper. We remain ever appreciative! We're sorry we didn't reply sooner. We were working on a computationally efficient version of IMM. We finally managed to make the formulation that we proposed in the general rebuttal work for the logistic regression example. We report on this in a comment to the general rebuttal. We hope that you will read it.
> > > > >
> > > > > Regarding your suggestion about timely larger-scale applications, we understand that in this day and age, it feels like any ML research that isn't directly tested in the latest language, vision, or robotics applications is not as applicable. However, for small academic research groups without an existing pipeline into these disciplines, especially PhD students working just with their advisors, it is often not feasible to implement such large-scale tests. Instead, we rely on smaller-scale problems that still have the essence of those larger problems, where success strongly suggests potential success if scaled. An analogy here would be a mechanical engineering researcher proposing a design change that could make combustion engines more efficient. We would likely all agree that this researcher should not be expected to build a whole engine and a car in order to demonstrate that their design idea is valid.
> > > > >
> > > > > Perhaps, however, you are not suggesting that we actually build a large language model with IMM, or anything of that scale, and that you rather only want us to evoke these potential applications. If that is what you mean, we have in fact several ideas where IMM could be very impactful. Here are two:
> > > > >
> > > > > 1. **Infusing Models with Physics**: We already suggested this to a fellow reviewer.  One setting we contemplated to try IMM in is in building physics-informed models. A closed-form physics model (e.g., climate prediction) can be thought of as a restricted model, because it often relies on very specific features. A full-featured model could use many more features than the physics model. IMM can be used as a method to incorporate the physics model, by making sure that the full-featured model's induced version matches the physics model.
> > > > > 2. **Augmenting Sensing**: This scenario was partly behind the RL example in the paper. Imagine an autonomous driving setting where a model is built using a certain set of sensors (e.g., depth camera and ultrasound sensors). The policy may have been trained using a lot of expensively collected imitation learning data, and it performs driving very well with just these sensors. What if we decide to add additional sensors (e.g., Lidar)? Do we need to re-collect all that driving data now with the new sensors on? Or can we use the previous model/policy as we train a new one that _also_ uses the Lidar? We hope that it is evident that this is a problem of a feature-restricted model informing a fully-featured one, and that thus IMM can play a big role in enabling it and saving the cost of collecting brand new data.
> > > > >
> > > > > (In a previous revision of this paper, we had these examples, but had to remove them due to space. In our conclusion, we also mention the potential usefulness of IMM as a means to extend the context window of LLMs.)
> > > > >
> > > > > We hope that these problems are thought-provoking and enough to convince you that the usefulness of IMM is not just in a few lines of math and basic experiments, and that there is merit to those ideas beyond the pages of the paper, especially to those who can more readily perform wider experiments and implementations. However, unless the paper is shared with the community, those ideas will remain not as publicized. We hope that you will help us share it.
> > > > >
> > > > > Thank you again for everything.
> > > > >
> > > > > Sincerely,
> > > > > Authors

---

### Author Rebuttal · Authors · 2024-08-07

We thank you all for your insightful and positive reviews. We are encouraged by your appreciation of our work and for your constructive criticism. We are lucky to have received such high quality feedback.

We have individually addressed all the points that you've raised. There is, however, one common theme that arose across the reviews: **whether IMM can be made more computationally efficient.**

+ While the paper itself is focused mostly on the statistical aspect of the problem, we recognize that scalability of IMM is critical for adoption. As such, we have been working on this issue, and believe that there is an elegant solution that could potentially achieve this scalability. Unfortunately, this work is not complete yet and we don't have experiments paralleling those in the paper. However, we are happy to share with you our insights.

+ The main bottleneck with IMM is the need to calculate the learned induced model. In the paper, we are doing this by sampling $k$ long histories and averaging the outputs. This has certain advantages, such as giving us a low-variance estimate of the gradient. However, it can be computationally expensive, especially in recurrent models which require new passes over the substituted histories.

+ The following alternative idea is similar to the sequentialization aspect covered in Appendix C.2, Equation (19). Using $\overline{x}$ to refer to short history and $\underline{x}$ to refer to long history, we can write the idealized IMM loss as ([see this figure for reference of this new streamlined notation](https://i.imgur.com/NwQgOPV.png)):

$ - \sum_\overline{x} \pi(\overline{x}) \sum_y \overline{P}(y|\overline{x}) \log \overline{Q}(y|\overline{x})$

where the learned induced model is:

$ \overline{Q}(y|\overline{x}) = \sum_\underline{x} \pi(\underline{x}|\overline{x}) Q(y|\underline{x},\overline{x})$.

The gradient of this IMM loss then becomes:

$ - \sum_\overline{x} \pi(\overline{x}) \sum_y \overline{P}(y|\overline{x}) \frac{\sum_\underline{x} \pi(\underline{x}|\overline{x}) \nabla Q(y|\underline{x},\overline{x})}{\overline{Q}(y|\overline{x})}$

$ = - \sum_\overline{x} \sum_\underline{x} \pi(\overline{x}) \pi(\underline{x}|\overline{x}) \sum_y \overline{P}(y|\overline{x}) \frac{ \nabla Q(y|\underline{x},\overline{x})}{\overline{Q}(y|\overline{x})}$

The empirical version of this gradient is:

$ - \frac{1}{n} \sum_t \sum_y \overline{P}(y|\overline{x}_t) \frac{ \nabla Q(y|\underline{x}_t,\overline{x}_t)}{\overline{Q}(y|\overline{x}_t)}$

As you can see, there is no sampling of any histories in this expression! It can even be rewritten in the form of cross-entropy, making the connection with noising and reverse-KD even more apparent, as it introduces a correction factor:

$ - \frac{1}{n} \sum_t \sum_y \overline{P}(y|\overline{x_t})  \underbrace{\frac{Q(y|\underline{x_t},\overline{x_t})}{\overline{Q}(y|\overline{x_t})}}_{\textsf{correction}} \nabla \log Q(y|\underline{x_t},\overline{x_t})$

The fascinating thing about this approach is that it only has a constant factor overhead on the baseline of cross-entropy training. There are two caveats, however:
+ This has a higher variance than the sampling approach of the paper. Reducing the variance is challenging, and we are investigating it using momentum-based approaches.
+ It requires the maintenance of an induced-model estimate. We can solve this relatively easily for the bigram case, by accumulating prediction vectors into a matrix.

We are sharing this insight with you in the hope of convincing you that there are indeed avenues to making IMM computationally effective. We don't know whether we'd be able to share with you parallel experiments by the end of the discussion period, but we hope that you will take this into consideration and become even more confident in the merits of this work. Thank you very much for all your time and effort!

*Note: The attached PDF and the links in the rebuttals are all anonymized and contain the same two figures.*

---

> ### Author Response · Authors · 2024-08-14
> **An important update on this computationally efficient version.**
>
> We have been hard at work at researching the computationally efficient implementation of the approach suggested above. Let's call it *no-sampling IMM*, to contrast it with the paper's *sampled IMM*. We were finally able to implement this approach in the logistic regression example in a satisfactory way, by addressing the two caveats mentioned above:
>
> + **High-variance**: The gradient of *no-sampling IMM* does not have the $k$-sample averaging and has thus higher variance. To compensate for this, we experimented with SGD with momentum. This helped most of the time, but unfortunately also frequently caused instability. We traced this instability to two sources: staleness in the correction factor, which we address in the second point, and exploding gradients. We controlled the latter using gradient-clipping. When using momentum, since the learning rate effectively increases, the optimization also sometimes got stuck in plateaus. We solved this by using learning-rate decay schedules.
>
> + **Maintaining an induced model**: In order not to compute the full induced model at every iteration (which would bring back the cost that we are trying to eliminate), we experimented with updating the model every $n$ iterations. By doing so, even if we recompute a full induced model using $\mathcal{O}(n^2)$ computation (which is the extreme of $k=n$, using _all_ samples in the density estimator, see Sec. 6.1), it will only cost $\mathcal{O}(n)$ additional computation, which is the complexity of the baseline. However, this causes the estimate of $\overline{Q}$ to become stale (since $Q$ gets updated between recomputations of $\overline{Q}$). Say model $Q_\dagger$ was used to calculate $\overline{Q_\dagger}$. We found that if we use the updated/fresh $Q$ for the correction factor, as in $\frac{Q}{\overline{Q_\dagger}}$, then it causes instability in the gradient, likely due to these correction factors not obeying expected constraints, e.g., they should have mean $1$ for every $y$, but they won't if one is fresh the other stale. This reasoning gave us the following solution: we found that if we use the stale model instead, i.e., if we use $\frac{Q_\dagger}{\overline{Q_\dagger}}$ for the correction factor, then the instability disappears.
>
> Here is a comparison of the average performance (over 20 runs) of *no-sampling IMM* and *sampled IMM*. The latter is as in the paper. The former uses the following hyperparameters: $\lambda$ is tuned the same way as for the sampled version, $\overline{Q}$ is updated every $n$ samples, momentum is set at 0.85, clipping is done with the inf-norm at threshold 2, a plateau-based learning rate scheduler attenuates learning by 0.9 at each plateau, the max number of epochs is set to 20,000, with early stopping based on convergence.
>
> This computationally efficient version continues to have the edge on noising. The match between the two versions improves with more data. We hypothesize that this is due to the fact that *no-sampling IMM* approximates the ideal IMM loss directly empirically, whereas *sampled IMM* does this through estimating $\overline Q$. In terms of clock time, *no-Sampling IMM* is considerably faster than *sampled IMM* and comparable to the baseline (we clocked it to take 50-70% more time at most than the baseline. (e.g., on a multicore node with a V100 GPU, a typical run for the *no-sampling IMM* vs. baseline is 4.5 vs. 2.8 seconds at size 10, and 20 vs. 12 seconds at size 50.)
>
> | Dataset Size |   Baseline           | *sampled IMM*            | *no-sampling IMM*                 |
> |--------------|----------------------|---------------------|---------------------|
> | 5          | 73.14  | 79.96 | 77.78  |
> | 10         | 84.17  | 88.65 | 86.42  |
> | 15         | 89.86  | 92.52 | 91.02  |
> | 20         | 92.35  | 94.30 | 95.27  |
> | 30         | 94.94  | 95.69 | 95.65  |
> | 40         | 96.33  | 96.59 | 96.52  |
>
> Finally, although we did not implement this in the language model case yet, we are confident that the observations we made will carry over (momentum, gradient clipping, learning rate schedules, intermittently updating $\overline{Q}$, using the stale model). Notably, to induce a bigram, one only needs to do a single pass through the data. This would give $\overline{Q_\dagger}$ . As for $Q_\dagger$, the most space-efficient way to access it is to keep a copy of that model in memory (the logits over the entire dataset would require more space to store.)
>
> ---
>
> Our paper was concerned primarily with the statistical aspects of the problem. We are very thankful to all of you for pushing us to think harder about making IMM also computationally attractive. We don't claim to have completely addressed it, however we **(1)** gave a high-level  formulation to ease the computation, and **(2)** demonstrated its validity in an example from the paper (albeit a toy one). We hope that you will take this earnest effort into consideration for your final recommendations. Thank you.

---

### Decision · Program_Chairs · 2024-09-25

**Decision:**

Accept (spotlight)

**Comment:**

This paper explores an interesting setting in which a (very accurate) model relying on a small subset of features is available as side information to a modeler who seeks to now train a larger fuller-featured model on a full dataset. The paper proposes a new approach called IMM, showing results on simple models like logistic regression and extending the evaluation to include results with deep nets and language models. Reviewers all vote for acceptance, finding the paper to be well-written, appreciating the connections to the knowledge distillation literature, appreciating the strength of the analysis, and finding the idea to be practical.